



# The role of air-sea fluxes for the water vapour isotope signals in the cold and warm sectors of extratropical cyclones over the Southern Ocean

Iris Thurnherr[1], Katharina Hartmuth[1], Lukas Jansing[1], Josué Gehring[2], Maxi Boettcher[1], Irina Gorodetskaya[3], Martin Werner[4], Heini Wernli[1], and Franziska Aemisegger[1]

[1]Institute for Atmospheric and Climate Science, ETH Zürich, Zurich, Switzerland
[2]Environmental Remote Sensing Laboratory, École Polytechnique Fédérale de Lausanne, Lausanne, Switzerland
[3]Centre for Environmental and Marine Studies, Department of Physics, University of Aveiro, Portugal
[4]Alfred Wegener Institute, Helmholtz Centre for Polar and Marine Research (AWI), Bremerhaven, Germany

**Correspondence:** Iris Thurnherr (iris.thurnherr@env.ethz.ch)

**Abstract.** Meridional atmospheric transport is an important process in the climate system and has implications for the availability of heat and moisture at high latitudes. Near-surface advection of cold and warm temperature over the ocean in the context of extratropical cyclones additionally leads to important air-sea exchange. In this paper, we investigate the impact of these air-sea fluxes on the stable water isotope (SWI) composition of water vapour in the Southern Ocean's atmospheric bound-

ary layer. SWIs serve as a tool to trace phase change processes involved in the atmospheric water cycle and, thus, provide important insight into moist atmospheric processes associated with extratropical cyclones. Here we combine a three-month ship-based SWI measurement data set around Antarctica with a series of regional high resolution numerical model simulations from the isotope-enabled numerical weather prediction model COSMO$_{iso}$. We objectively identify atmospheric cold and warm temperature advection associated with the cold and warm sector of extratropical cyclones, respectively, based on the air-sea

temperature difference applied to the measurement and the simulation data sets. A Lagrangian composite analysis of cold and warm temperature advection based on the COSMO$_{iso}$ simulation data is compiled to identify the main processes affecting the observed variability of the isotopic signal in marine boundary layer water vapour in the region from $35°$ S to $70°$ S . This analysis shows that the cold and warm sectors of extratropical cyclones are associated with contrasting SWI signals. Specifically, the measurements show that the median values of $\delta^{18}$O and $\delta^{2}$H in the atmospheric water vapour are 3.6 ‰ and 23.2 ‰ higher

during warm than during cold advection. The median value of the second-order isotope variable deuterium excess $d$, which can be used as a measure of non-equilibrium processes during phase changes, is 5.9 ‰ lower during warm than during cold advection. These characteristic isotope signals during cold and warm advection reflect the opposite air-sea fluxes associated with these large-scale transport events. The trajectory-based analysis reveals that the SWI signals in the cold sector are mainly shaped by ocean evaporation. In the warm sector, the air masses experience a net loss of moisture due to dew deposition as

they are advected over the relatively colder ocean, which leads to the observed low $d$. We show that additionally the formation of clouds and precipitation in moist adiabatically ascending warm air parcels can decrease $d$ in boundary layer water vapour. These findings illustrate the highly variable isotopic composition in water vapour due to contrasting air-sea interactions during





cold and warm advection, respectively, induced by the circulation associated with extratropical cyclones. SWIs can thus potentially be useful as tracers for meridional air advection and other characteristics associated with the dynamics of the storm
tracks over interannual timescales.

## 1    Introduction

Ocean evaporation is the most important source of atmospheric water vapour and it impacts mid-latitude and polar atmospheric and ocean dynamics. Strong ocean evaporation can lead to the intensification of extratropical cyclones (e.g. Yau and Jean, 1989; Uotila et al., 2011; Kuwano-Yoshida and Minobe, 2016) and polar lows (Rasmussen and Turner, 2003) and to changes
in atmospheric (Neiman et al., 1990; Sinclair et al., 2010) and ocean static stability, for example in cyclone-induced cold ocean wakes (Chen et al., 2010) or by inducing deep water formation at high latitudes (Condron et al., 2006; Condron and Renfrew, 2013). The strength of ocean evaporation in the extratropics is strongly modulated by the large-scale atmospheric flow. Ocean evaporation averaged across extratropical cyclones is similar to the ocean evaporation in the North Atlantic (Rudeva and Gulev, 2010) and the Southern Ocean (Papritz et al., 2014). But there are large differences in ocean evaporation between the cold and
warm sectors of extratropical cyclones, which are regions within cyclones of equatorward and poleward air mass transport, respectively. The equatorward advection of dry and cold air in the cold sector of extratropical cyclones leads to a large air-sea moisture gradient and strong large-scale ocean evaporation (Bond and Fleagle, 1988; Boutle et al., 2010; Aemisegger and Papritz, 2018), while weak ocean evaporation or even moisture fluxes from the atmosphere to the ocean, i.e. dew deposition, are observed ahead of the cold front in the warm sector (Fleagle and Nuss, 1985; Persson et al., 2005; Bharti et al., 2019). In
polar regions close to the sea ice edge, cyclones induce the advection of cold and dry air over the open ocean leading to cold air outbreaks and strong ocean evaporation (Papritz et al., 2015). Thus, the large-scale meridional advection modulates air-sea interactions, especially in the storm track regions.

The opposite direction of surface fluxes in the cold and warm sector of extratropical cyclones impact the marine boundary layer (MBL) stability and moisture budget. In the cold sector, positive sensible heat fluxes lead to a low atmospheric stability and a
high MBL height (Beare, 2007; Sinclair et al., 2010). In idealised model simulations, the strongest ocean evaporation is seen directly behind the cold front in the region of subsiding dry air (Boutle et al., 2010). Dew deposition on the ocean surface has been observed ahead of the cold front (Persson et al., 2005) and in warm sectors during the passage of extratropical cyclones over cold ocean regions (Neiman et al., 1990). Negative sensible heat fluxes in the warm sector lead to a high boundary layer stability and shallow MBLs (Beare, 2007; Sinclair et al., 2010). The net freshwater flux between the ocean and atmosphere is
the sum of ocean evaporation and precipitation. In the warm sector, important precipitation occurs in the region of the warm conveyor belt - a moist, coherent, ascending airstream in front of the cold front (Browning, 1990; Madonna et al., 2014; Pfahl et al., 2014). Frontal precipitation, often related to the warm conveyor belt, can affect the surface moisture fluxes in both sectors of extratropical cyclones (Catto et al., 2012).

The characteristic surface freshwater fluxes in the cold and warm sector of extratropical cyclones are schematically summarised
in Fig. 1. Air-sea moisture fluxes in the form of ocean evaporation and dew deposition are caused by a thermodynamic dis-





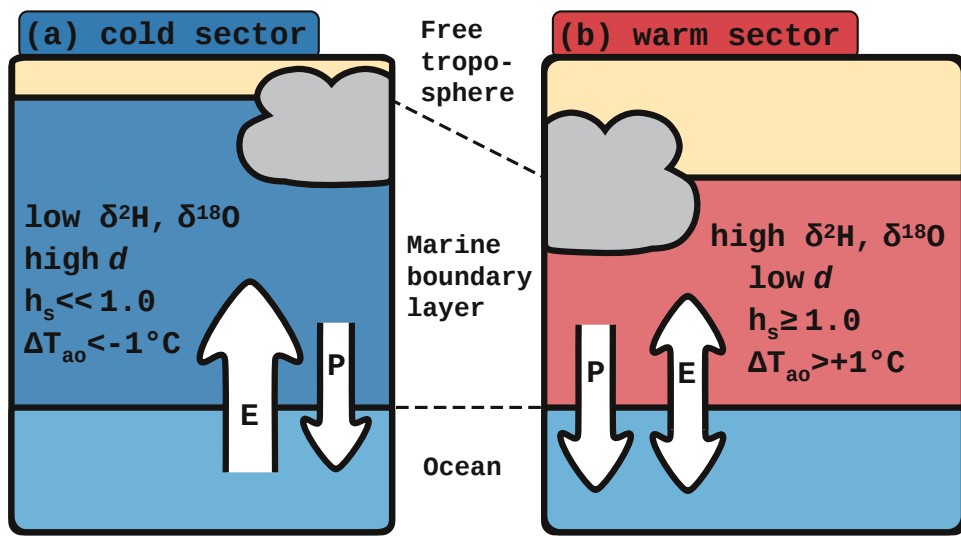

**Figure 1.** Schematic of air-sea interactions in **(a)** the cold and **(b)** the warm sector of an extratropical cyclone. $E$ denotes ocean evaporation and, if directed downward, dew formation and $P$ denotes precipitation. $h_s$ is the relative humidity with respect to sea surface temperature and $\Delta T_{ao}$ the difference between the air and sea surface temperature. For details see text.

equilibrium between the ocean and atmosphere, which can be expressed by the relative humidity with respect to sea surface temperature $h_s = \frac{q_a}{q_s(SST)}$, where $q_a$ is the specific humidity of the atmosphere and $q_s(SST)$ the saturation specific humidity at sea surface temperature (SST). $q_s$ is temperature-dependent and increases with increasing temperature $T_a$. In the cold sector, where dry and cold air is advected over a relatively warm ocean surface, the air-sea temperature difference $\Delta T_{ao} = T_a - SST$

is negative and $h_s$ is low due to a low $q_a$ and a relatively high $q_s$. Therefore, the air in the cold sector is generally undersaturated with respect to the ocean and, thus (intense) ocean evaporation is expected (Fig. 1a). Negative surface freshwater fluxes can be observed in the cold sector due to precipitation. The horizontal advection in the cold sector of extratropical cyclones is referred to as cold temperature advection in the following. In contrast, in the warm sector, warm air is advected over a relatively colder ocean surface (referred to as warm temperature advection in the following). Under these environmental conditions,

$\Delta T_{ao}$ is positive and $h_s$ is high due to a high $q_a$ relative to $q_s$. Therefore, the near-surface air in the warm sector can be close to saturation or oversaturated with respect to the potentially cold ocean surface, leading to weak ocean evaporation or dew deposition (Fig. 1b). Furthermore, precipitation associated with the warm conveyor belt leads to moisture fluxes from the atmosphere to the ocean.

Despite their important role in the atmospheric moisture budget, only few and regionally limited measurements of ocean

evaporation and dew deposition are available, because of the extensive set of measurements needed (Pollard et al., 1983; Fleagle and Nuss, 1985; Holt and Raman, 1990; Neiman et al., 1990; Persson et al., 2005; Bharti et al., 2019). Many studies on air-sea moisture fluxes in extratropical cyclones rely on model simulations (e.g. Nuss, 1989; Beare, 2007; Sinclair et al., 2010; Boutle





et al., 2011). Further insight into the strength of ocean evaporation can be gained by ship-based measurements of humidity and of stable water isotopes (SWIs) in water vapour, which provide near-surface water vapour characteristics. SWI measurements

can be used to better understand the importance of various moist processes including ocean evaporation, dew deposition and precipitation for the MBL moisture budget. The relative abundance of heavy and light isotopes in the different water reservoirs is altered during phase-change processes due to isotopic fractionation. The abundance of the heavy isotopologues $^2$H$^1$H$^{16}$O and $^1$H$_2^{18}$O is expressed by the $\delta$-notation ($\delta^2$H and $\delta^{18}$O, respectively) (Dansgaard, 1964), which is defined as the isotopic ratio $R$ of the concentration of the heavy isotopologue to the concentration of the light isotopologue (hereafter named isotope)

$^1$H$_2^{16}$O relative to an internationally accepted standard isotopic ratio (the Vienna standard mean ocean water, VMSMOW2; with $^2R_{\text{VSMOW2}}$=1.5576·10$^{-4}$ and $^{18}R_{\text{VSMOW2}}$=2.0052·10$^{-3}$): $\delta^2$H$[\permil]=(\frac{^2R_{sample}}{^2R_{\text{VSMOW2}}\cdot 2}-1)\cdot 1000$ and $\delta^{18}$O$[\permil]=(\frac{^{18}R_{sample}}{^{18}R_{\text{VSMOW2}}}-1)\cdot 1000$. There are two types of isotopic fractionation: equilibrium fractionation, which is caused by the difference in saturation vapour pressure of different isotopes, and non-equilibrium fractionation, which occurs due to molecular diffusion, e.g. during ocean evaporation. A measure of non-equilibrium fractionation and, thus, diffusive processes such as evaporation or dew

deposition is the second-order isotope variable deuterium excess $d$, defined as $d=\delta^2$H$-8\cdot\delta^{18}$O. During a diffusive process, a positive anomaly in $d$ develops in the moisture sink (e.g. the atmosphere during ocean evaporation), while a negative $d$ anomaly can be observed in the moisture reservoir (e.g. rain droplets during below-cloud evaporation). If the moisture reservoir is large and well-mixed, which can e.g. be assumed for the ocean, the impact of isotopic fractionation on the isotopic composition of the reservoir can be neglected. Air-sea net moisture fluxes occur due to non-equilibrium conditions at the atmosphere-ocean

interface and, therefore, $d$ can be used as a tracer of air-sea interactions.

Previous studies have shown, that $d$ in MBL water vapour negatively correlates with $h_s$ (Uemura et al., 2008; Pfahl and Wernli, 2008; Bonne et al., 2019; Thurnherr et al., 2020), which reflects the differing strength of ocean evaporation and, thus, non-equilibrium fractionation in different $h_s$-environments. So far, studies focused on environments with low $h_s$, where positive $d$ in atmospheric water vapour has been observed due to strong ocean evaporation (Gat et al., 2003; Uemura et al., 2008; Gat,

2008; Pfahl and Wernli, 2008; Aemisegger and Sjolte, 2018). In extratropical cyclones, such low $h_s$-environments with high $d$ are expected in the cold sector, corresponding to areas of strong large-scale ocean evaporation (Aemisegger and Sjolte, 2018, and Fig. 1a). An opposite signal in $d$, i.e. $d$ close to or below 0, is expected in the warm sector, where dew deposition or weak ocean evaporation occurs (Fig. 1b). Only few measurements of negative $d$ in atmospheric water vapour have been reported (Uemura et al., 2008; Bonne et al., 2019; Thurnherr et al., 2020), which have been related to weak ocean evaporation (Uemura

et al., 2008) and deposition of water vapour on sea ice (Bonne et al., 2019). However, to our current knowledge there is no study that investigated the processes leading to the $d$-$h_s$ relationship in MBL water vapour in high $h_s$-environments.

Modelling of SWIs in the atmospheric branch of the water cycle helps to identify which moist processes influence the isotopic composition of water vapour. The incorporation of SWIs into numerical climate and weather models (Joussaume et al., 1984; Yoshimura et al., 2008; Blossey et al., 2010; Risi et al., 2010; Werner et al., 2011; Pfahl et al., 2012) allows to study the impact of

different moist atmospheric processes on the SWI-evolution of atmospheric water vapour. Such model simulations also provide a spatial context to measurement data and the basis for the definition of various useful Eulerian and Lagrangian diagnostics. Recently, in situ SWI observations have been used to compare the representation of the hydrological cycle in general circulation





models equipped with water isotopes, showing large differences in the performance of the 3 atmospheric general circulation models ECHAM-wiso, LMDZiso and isoGSM (Steen-Larsen et al., 2017). Detailed insights into the interaction of weather

systems and SWIs can be obtained from the isotope-enabled Consortium for Small-Scale Modelling model COSMO$_{iso}$ (Pfahl et al., 2012). Lagrangian studies based on COSMO$_{iso}$ simulations have shown that $\delta^2$H and $d$ in near-surface water vapour are strongly influenced by ocean evaporation and, over land, by evapotranspiration and mixing with moist air, while liquid and mixed phase cloud formation contributes to the $\delta^2$H- and $d$-variability (Aemisegger et al., 2015; Dütsch et al., 2018). The importance of different air mass origins and pathways of airstreams for the SWI evolution during frontal passages has been

illustrated using a COSMO$_{iso}$ simulation in the Mediterranean Sea (Lee et al., 2019). Idealised COSMO$_{iso}$ simulations of an extratropical cyclone showed that the meridional advection of air in the cold and warm sector strongly shape the characteristic high $\delta$ values in the warm and low $\delta$ values in the cold sector (Dütsch et al., 2016, see also Fig. 1). In their idealised model study, isotopic fractionation during ocean evaporation was switched off, such that air-sea interaction processes did not affect their simulated $\delta^2$H-contrast between the cold and warm sector. It is, therefore, not known yet how important air-sea interactions

are in shaping the high $\delta$ values in the warm sector and the low $\delta$ values in the cold sector of extratropical cyclones (see also Fig. 1).

In this study, we aim to address the following questions analysing three-month ship-based SWI measurements in the Southern Ocean in combination with high resolution regional COSMO$_{iso}$ simulations covering the measurement period with the goal to better understand the influence of air-sea interactions on the isotopic composition of water vapour in the MBL:

1. What is the occurrence frequency of cold and warm temperature advection over the Southern Ocean?

    2. What characteristic SWI signal is measured in the cold and warm sector of extratropical cyclones, respectively?

    3. How do the differing air-sea freshwater fluxes in the cold and warm sectors of extratropical cyclones affect the isotopic variability of the MBL water vapour?

This paper is structured in the following way: In Sect. 2, the measurements, simulations and Lagrangian diagnostics are de-

scribed. An objective identification of cold and warm temperature advection is introduced in Sect. 3. Thereafter, the temperature advection climatology is discussed (Sect. 4.1), the measured SWI signals are related to the cold and warm temperature advection in the cold and warm sector of extratropical cyclones, respectively (Sect. 4.2), and the evolution of the isotope signals along COSMO$_{iso}$ trajectories is analysed for trajectories from the cold and warm sector, respectively (Sect. 4.3).

## 2 Data

This study combines meteorological and SWI measurements from the Antarctic Circumnavigation Expedition (ACE), which took place from 21 December 2016 to 19 March 2017 in the Southern Ocean (Walton and Thomas, 2018; Schmale et al., 2019, Sect. 2.1), with COSMO$_{iso}$ simulations and ERA-Interim reanalysis data from the European Centre for Medium-Range Weather Forecasts (ECMWF) (Sect. 2.2) . The Lagrangian methods used for the analysis are described in Sect. 2.3.



## 2.1 Measurement data

Different measurement data sets from ACE were used in this study:

1. During ACE (see cruise track in Fig. 2), continuous measurements of SWIs using a Picarro cavity ring-down laser spectrometer were conducted at a height of 13.5 m a.s.l. above the ocean surface on board the Russian research vessel *Akademik Tryoshnikov*. A detailed description of the SWI data set (setup and post-processing) can be found in Thurnherr et al. (2020).

2. A merged SST product (Haumann et al., 2020), that is a combination of in situ measurements using an Aqualine Ferrybox system and, where no in situ measurements are available, of the daily optimum interpolation SST satellite product from the Advanced Very High Resolution Radiometer (AVHRR) infrared sensor (version 2; AVHRR-Only; Reynolds et al., 2007).

3. Air temperature, air pressure and relative humidity are used from the automated weather station operated onboard
(Landwehr et al., 2019).

4. Rainfall and snowfall rates along the ship track are derived from continuous micro rain radar (MRR) measurements. The rainfall rate (Gehring et al., 2020) is computed from the drop size distribution estimated from the MRR Doppler spectra at the 100-200 m a.s.l. range gate as explained in Peters et al. (2005). During time periods with a melting layer close to the surface, the rainfall rate is strongly overestimated by our method. These periods of a low melting layer (<200 m a.s.l.)
are masked for the analysis in this study. The snowfall rates are calculated from the MRR effective reflectivity at the 400 m a.s.l. range. The effective reflectivity along with the Doppler velocity and the spectral range are derived from the raw Doppler spectra using the algorithm of Maahn and Kollias (2012). In order to estimate the snowfall rate, we use the Z-S relationship derived by Grazioli et al. (2017) based on the measurements at Dumont-D'Urville station. We assume that the snowfall measurements at this location (on an island near Adelie Land) provide a good approximation of the snowfall
microphysical properties observed during ACE.

5. To study the vertical structure of the MBL, radiosonde measurement are used. iMET radiosondes were deployed during ACE measuring pressure, air temperature, relative humidity and the GPS location. The radiosondes were launched once or twice per day and at higher frequency for specific events.

All data is used at an hourly time resolution, except for the radiosondes which were launched at specific times with vertical
profile measurements available at 1 s resolution.

## 2.2 Model data

### 2.2.1 ERA-Interim reanalysis data

Six-hourly data from the ERA-Interim reanalyses (Dee et al., 2011) spanning the time period from 1979 to 2018 are used for the climatological analysis of warm and cold temperature advection (see Sect. 3) over the Southern Ocean. The data is interpolated

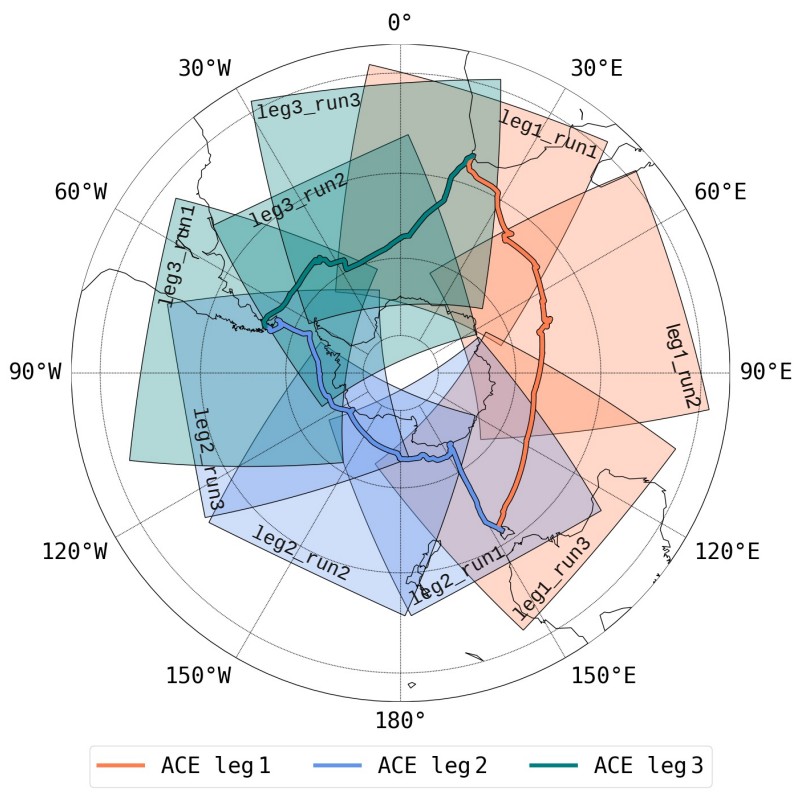

**Figure 2.** Domains of the nine COSMO$_{iso}$ simulations (see also Table 1). The ACE ship track is indicated by a bold line, coloured differently for the three legs of the expedition.

to a 1° horizontal grid and 60 vertical levels. Due to fewer observational data in the Southern Hemisphere compared to the Northern Hemisphere, the model fields are expected to have a higher uncertainty in the Southern Hemisphere. Nonetheless, the 4D-var data assimilation system by the ECMWF to compile the ERA-Interim reanalysis shows good performance in the Southern Hemisphere (Dee et al., 2011; Nicolas and Bromwich, 2011) and the uncertainties due to assimilation errors have been shown to be minor (Nicolas and Bromwich, 2011).

Based on the ERA-Interim reanalyses, cyclone frequencies were calculated using a 2D cyclone detection algorithm based on sea level pressure fields (Wernli and Schwierz, 2006; Sprenger et al., 2017). Furthermore, surface fronts were detected following Schemm et al. (2015) using the following criteria: (i) the horizontal equivalent potential temperature gradient at 850 hPa has to be at least 3.8 K (100 km)$^{-1}$, and (ii) the fronts need to have a minimum length of 500 km.

### 2.2.2   COSMO$_{iso}$ simulations

The limited-area regional numerical weather prediction model COSMO (Steppeler et al., 2003) is used in its isotope-enabled version COSMO$_{iso}$ (Pfahl et al., 2012) with two additional parallel water cycles for the heavy water molecules H$_2^{18}$O and



$^1$H$^2$H$^{16}$O, which mirror the water cycling of H$_2^{16}$O. These additional water cycles are affected by the same physical processes as the light water molecule, except for isotopic fractionation during phase change processes. COSMO$_{iso}$ has been previously used to study various aspects of the regional atmospheric water cycle over Europe and the USA (Pfahl et al., 2012; Aemisegger et al., 2015; Dütsch et al., 2018; Christner et al., 2018; Lee et al., 2019). Here, nine COSMO$_{iso}$ simulations were conducted in the Southern Ocean covering the ACE measurement time period. The simulations were initialised and driven at the lateral boundaries by output from an ECHAM5-wiso simulation, which was nudged 6-hourly to temperature, surface pressure, divergence and vorticity of ERA-Interim reanalysis data (Werner et al., 2011; Butzin et al., 2014). The wind in the COSMO domain was spectrally nudged to the ECHAM5-wiso above 850 hPa to keep the meteorology within the COSMO$_{iso}$ domain as close as possible to the reanalysis. The ECHAM5-wiso fields are available 6-hourly with a spectral resolution of T106 (corresponding to 88 km at 45° S) and 31 vertical levels. The COSMO$_{iso}$ simulations were performed at a horizontal resolution of 0.125°, corresponding to ∼14 km, with 40 vertical levels and explicit convection. The explicit convection setup is preferred over using the deep convection parametrisation even at a resolution of 0.125°. Simulations with the COSMO model over Europe have been shown to represent the hydroclimate more realistically in this setup (Vergara-Temprado et al., 2019). Furthermore, isotope-enabled simulations over the Southern Ocean with explicit convection revealed a reduced strength of vertical mixing and more realisitic vertical isotope profiles than simulations with parametrised convection (not shown, see also results from Jansing, 2019, with parametrised convection). Hourly outputs of the COSMO$_{iso}$ simulations are used for the analysis. The model domains have an area of approximately 50° x 50° and were shifted along the ACE track, such that the entire expedition route was covered by the 9 simulations in space and time (see Fig. 2). Isotopic fractionation during surface evaporation is parameterized with the Craig–Gordon model (Craig and Gordon, 1965) using a wind speed independent formulation of the non-equilibrium fractionation factor (Pfahl and Wernli, 2009). The isotopic composition of the ocean surface water is prescribed at a constant value of 1‰ for $\delta^2$H and $\delta^{18}$O, which is relatively high for the southern part of the ACE cruise track (Xu et al., 2012), but we expect that this has minor effects for the purposes of this study. For a detailed description of the physics and isotope parameterisations in the COSMO$_{iso}$ model see Doms et al. (2013) and Pfahl et al. (2012), respectively. For terrestrial surfaces, a one-layer surface snow model with equilibrium fractionation during snow sublimation and a multi-layer soil model (see supplement of Christner et al., 2018) is used. The specifications of each model run are summarised in Table 1. The first week of each run is used as spin-up time and is not included in the analysis. For comparison with the ACE measurements, the lowest model level of the different variables is bi-linearly interpolated along the ACE track. This corresponds approximately to the height of the inlet on the ship. During time periods, when two model runs overlap in time and space, one of the runs is chosen in such a way that the individual cold and warm temperature advection events are extracted entirely from one single model run.

### 2.3 Backward trajectories and moisture sources

Seven-day air parcel backward trajectories are calculated using the Lagrangian analysis tool LAGRANTO (Wernli and Davies, 1997; Sprenger and Wernli, 2015) based on the three-dimensional 1-hourly wind fields from the COSMO$_{iso}$ simulations. The trajectories were launched every hour along the ACE track at pressure levels between 1000 and 500 hPa in 10 hPa-steps. Tra-




**Table 1.** Specifications of COSMO$_{iso}$ model runs. The columns center and width refer to the center and width of the simulation domain. The column time window shows the time period over which trajectories were calculated for the respective simulation.

| simulation | start [at 0 UTC] | end [at 0 UTC] | center [° S, ° E] | width [° lat, ° lon] | time window |
|---|---|---|---|---|---|
| leg1_run1 | 13 Dec 2016 | 12 Jan 2017 | 47, 18 | 50, 50 | 21 Dec 2016 - 1 Jan 2017 |
| leg1_run2 | 24 Dec 2016 | 23 Jan 2017 | 47, 73 | 50, 50 | 2 - 12 Jan 2017 |
| leg1_run3 | 3 Jan 2017 | 2 Feb 2017 | 52, 130 | 50, 50 | 13 - 26 Jan 2017 |
| leg2_run1 | 12 Jan 2017 | 11 Feb 2017 | 61, 151 | 50, 50 | 27 Jan - 8 Feb 2017 |
| leg2_run2 | 28 Jan 2017 | 27 Feb 2017 | 61, -154.4 | 50, 50 | 9 - 14 Feb 2017 |
| leg2_run3 | 1 Feb 2017 | 3 March 2017 | 61, -100 | 50, 50 | 15 - 23 Feb 2017 |
| leg3_run1 | 16 Feb 2017 | 18 March 2017 | 52, -80 | 47.5, 56.25 | 24 Feb - 1 March 2017 |
| leg3_run2 | 21 Feb 2017 | 23 March 2017 | 62, -25 | 50, 50 | 2 - 13 March 2017 |
| leg3_run3 | 1 March 2017 | 31 March 2017 | 47, -5 | 50, 50 | 14 - 23 March 2017 |

jectories were analysed until they left the model domain and for the time windows of each model run as indicated in Table 1. Several variables were interpolated along the trajectory positions, including the SWI concentrations such that the evolution of the SWI composition during the air mass transport can be analysed. Only trajectories starting within the MBL are used for the trajectory analysis in Sect. 4.3.

Moisture sources of the MBL water vapour along the ACE track were calculated using the moisture source diagnostic developed by Sodemann et al. (2008) adjusted to identify the moisture sources of water vapour (Pfahl and Wernli, 2008) using the seven-day COSMO$_{iso}$ backward trajectories in a setup as in Aemisegger et al. (2014). For the COSMO$_{iso}$ analyses in Sect. 4, only trajectories, for which at least 75% of the moisture upon arrival can be explained by moisture uptakes along the trajectories, are used. This corresponds to trajectories covering 83% of the ACE time period and explaining the origin of, on average,
89% of the moisture upon arrival.

The entrance time of the trajectories into a cold or warm sector before arrival at the measurement site is defined as the time when the trajectory enters the cold or warm sector (identified as explained in Sect. 3) without leaving the cold or warm sector, respectively, afterwards for more than 12 hours before arrival. The 12 h criteria is used to avoid that the entrance time is affected by short residence times outside of the sector.


## 3 Objective identification of cold and warm temperature advection

During the meridional advection of air masses in the cold and warm sector of extratropical cyclones, temperature advection occurs due to cold air, that is advected equatorward, and warm air, that is transported poleward. Temperature advection is



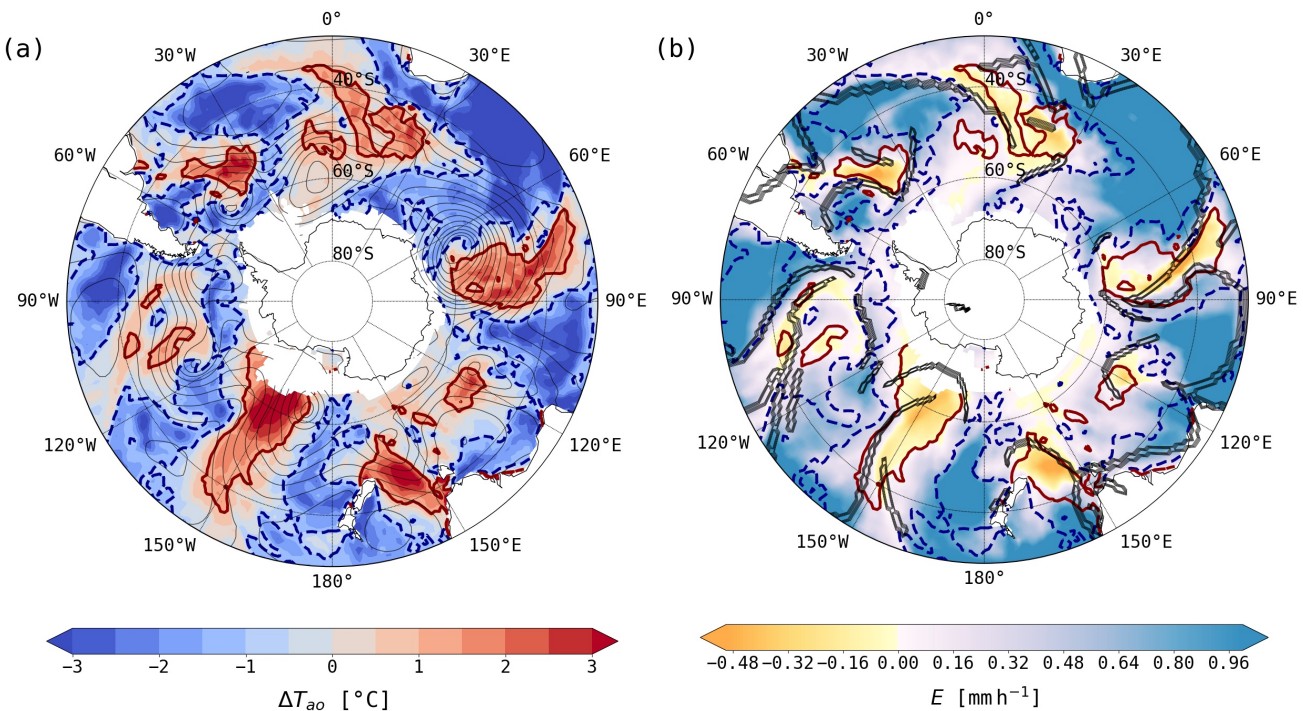

**Figure 3.** An example of cold and warm temperature advection events in the Southern Ocean at 12 UTC 26 Dec 2016 from ERA-Interim. **(a)** Air-sea temperature difference $\Delta T_{ao}$ using $T$ at 10 m a.s.l. (colours). Grey contours show sea level pressure in 5 hPa intervals. **(b)** Ocean evaporation $E$ (colours) and surface fronts (black contours). Contours of $\Delta T_{ao} = -1.0°$ C (blue, dashed) indicate cold and of $\Delta T_{ao} = 1.0°$ C (red, solid) warm temperature advection events, respectively. Land and areas covered by sea ice are blanked.

defined as $-\boldsymbol{u} \cdot \boldsymbol{\nabla} T$, where $\boldsymbol{u}$ is the velocity vector and $T$ the air temperature. To calculate $-\boldsymbol{u} \cdot \boldsymbol{\nabla} T$, the spatial distribution of

$T$ is needed, which is usually not available from ship-based meteorological measurements. Since the advection of air masses leads to a thermal imbalance between the ocean and the atmosphere, we use a simple identification method of temperature advection based on the air-sea temperature difference $\Delta T_{ao} = T - SST$, where $T$ is taken at a suitable near-surface level. A specific threshold of $\Delta T_{ao}$ is chosen to define cold and warm temperature advection, respectively. Cold temperature advection is defined as time periods when $\Delta T_{ao} < -1.0°C$ and warm temperature advection when $\Delta T_{ao} > 1.0°C$. A zonal or weak

advection regime is defined for $-1.0°C < \Delta T_{ao} < 1.0°C$. The effects of using different temperature thresholds to define the temperature advection regimes are discussed in Hartmuth (2019).

   Two-dimensional masks of cold and warm temperature advection events as identified by the proposed scheme in ERA-Interim using T at 10 m a.s.l. are shown exemplarily at 12 UTC 26 Dec 2016 in Fig. 3. The warm temperature advection masks cover areas to the north-east of low pressure systems (indicated by minima in sea level pressure), which correspond to the warm sectors of extratropical cyclones in the Southern Ocean (Fig. 3a). West of the low pressure systems, cold temperature advection in the






cold sectors can be seen. For example around 60° E, the cold and warm sectors of a large low pressure system are indicated by the cold and warm temperature advection masks. In this snapshot, areas of cold and warm temperature advection coincide with positive and negative ocean evaporation, respectively (Fig. 3b). Cold temperature advection is associated with strong ocean evaporation. Weak ocean evaporation occurs mainly between the advection masks, and very small or even negative moisture

fluxes indicate dew deposition occurring in the warm sectors (see for example at 40° W in Fig. 3b). As expected, the surface fronts often mark the boundaries between the cold and warm sectors and, thus, of the cold and warm temperature advection masks. The warm and cold fronts of the cyclone at 40° W delimit the warm sector along its southern edge following closely the temperature advection mask. In other cases, the surface fronts are not aligned with the temperature advection mask. This is the case for the cyclone at 60° E, where the cold front is within the warm temperature advection mask and negative ocean evapora-

tion is seen behind the cold front. This discrepancy between the surface fronts and the temperature advection masks could be caused by differences in the identification schemes. The surface fronts are identified using horizontal gradients in equivalent potential temperature at 850 hPa, while the advection mask is based on the contrast between $T$ at 10 m a.s.l and SST. The focus on air-sea interactions in this study justifies the choice of an identification scheme based on surface fields. Ocean evaporation aligns well with the advection masks confirming that the proposed identification scheme is useful for the investigation

of air-sea fluxes. This scheme is a simple objective method and can be applied to model simulations as well as measurement data. Other Eulerian features of extratropical cyclones, such as the cyclone centres, areas or fronts, have been identified using automated identification schemes (e.g. Lambert, 1988; Hewson, 1998; Wernli and Schwierz, 2006; Jenkner et al., 2010) and used to characterise the impact of extratropical cyclones on air-sea interactions (Papritz et al., 2014; Aemisegger and Papritz, 2018). The temperature advection scheme presented here provides the possibility to study the contrasting behaviour of air-sea

interactions specifically in the cold and warm sectors of extratropical cyclones, respectively.

In this study, the cold and warm temperature advection scheme was applied (i) to the ACE measurements, which are used for the characterisation of the isotopic signal in the cold and warm sectors, using the measured air temperature at 24 m a.s.l, and the merged SST product (see Sect. 2.1), (ii) to the ERA-Interim reanalysis, which is used for a cold and warm temperature advection climatology, using $T$ at 10 m a.s.l., and (iii) to the COSMO$_{iso}$ data set, which is used to study the relevant processes

shaping the SWI signal in cold and warm sectors, using $T$ at the lowest model level, which corresponds to approximately 10 m a.s.l. The lower level of air temperature used for the calculation of the COSMO$_{iso}$ and ERA-Interim advection events might lead to slightly lower frequencies compared to the ACE measurements, because larger temperature differences are expected across larger vertical distances. Nonetheless, the difference between 10 m and 24 m a.s.l. air temperature is fairly small and the advection frequencies in COSMO$_{iso}$ and ERA-Interim are similar to the advection frequencies in the measurements.

The largest difference in the identification of cold and warm temperature advection between the measurements and COSMO$_{iso}$ is observed during leg 2 (compare orange trajectories in Fig. 4 and black dots in Fig. 5). Two warm temperature advection events are identified during leg 2 in COSMO$_{iso}$, which were categorised as zonal flow using the measurements. During these two events, air is advected northwards from Antarctica towards the ship's position. The large positive $\Delta T_{ao}$ in COSMO$_{iso}$ could be caused by adiabatic warming during the descent in a katabatic wind event. These two warm temperature advection

events thus differ from a typical warm temperature advection event as generally observed along the ACE track in the warm



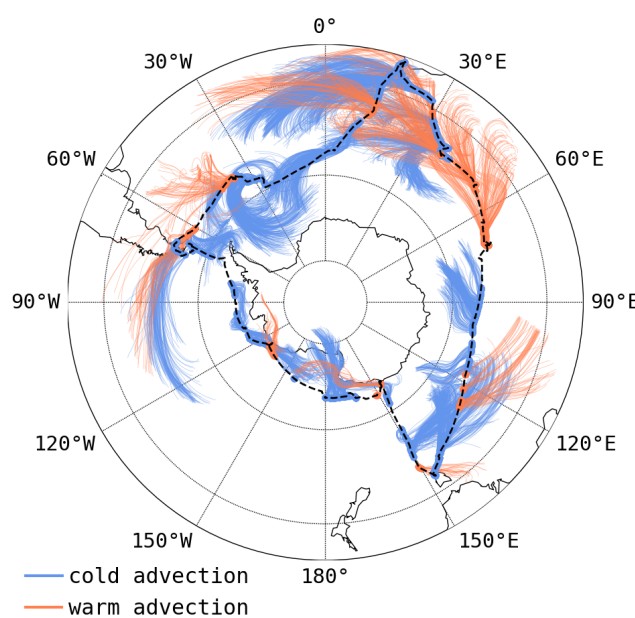

**Figure 4.** Two-day backward trajectories calculated with COSMO$_{iso}$ wind fields for warm (orange) and cold (blue) temperature advection events as identified using $\Delta T_{ao}$ in COSMO$_{iso}$ along the ACE ship track (black, dashed line).

sector of an extratropical cyclone. Overall, the identified cold and warm temperature advection events using COSMO$_{iso}$ agree well with the measurements and represent similar environmental conditions (see also Sect. 4.2). The COSMO$_{iso}$ backward trajectories along the ACE track show further that the identified cold and warm temperature advection events generally refer to situations of northward and southward flow, respectively (Fig. 4). We conclude, that the proposed identification scheme is

adequate to study the impact of meridional air mass advection on air-sea moisture fluxes in measurements and simulations.

# 4 Results

## 4.1 Frequencies of occurrence of cold temperature advection, warm temperature advection and zonal flow over the Southern Ocean

**a) Climatology for the Southern Ocean**

In order to characterise the temperature advection regimes climatologically, their occurrence frequency and associated air-sea moisture fluxes are analysed in the region south of 30° S for the period from December to March 1979 - 2018. Cold temperature advection, warm temperature advection and zonal flow occur with different climatological frequencies south of 30° S (Fig. 5).





Zonal flow is the most frequently occurring advection regime (49 %). Cold and warm temperature advection account for 39%
and 12 %, respectively. Each temperature advection regime, thus, occurs frequently and represents an important large-scale
flow situation of the atmospheric dynamics over the Southern Ocean. In the following, the large-scale flow environment and
freshwater fluxes climatologically associated with cold temperature advection, warm temperature advection and zonal flow are
discussed separately.

Cold temperature advection occurs during the meridional transport of cold air over a relatively warmer ocean surface in the
cold sector of extratropical cyclones. High occurrence frequency of cold temperature advection of up to 60% is seen in a
latitudinal band north of 40° S , equatorward of regions with high cyclone frequencies in all three ocean basins (Fig. 5a). In
these areas, the cold sectors of extratropical cyclones pass over regions with anomalously warm SSTs, that are higher than the
zonal mean SST (Fig. 6). For instance in the southeastern Indian Ocean, the two zonal SST maxima at 20° E and 60° E north of
40° S overlap with the local frequency maxima of cold temperature advection. In these regions, hot spots of large-scale ocean
evaporation occur frequently and are associated with the warm ocean western boundary currents along the continents (Moore
and Renfrew, 2002; Aemisegger and Papritz, 2018). Cold temperature advection also frequently occurs along the Antarctic
coast in the Ross Sea, Weddell Sea and across the Amery Ice shelf. These areas correspond to regions of frequent cold air
outbreaks in summer (Papritz et al., 2015). During cold air outbreaks, which are often induced by extratropical cyclones, cold
and dry air is advected over a relatively warm ocean. In the same regions along the Antarctic coast, strong large-scale ocean
evaporation events occur, of which more than 80% are driven by extratropical cyclones (Aemisegger and Papritz, 2018). Strong
evaporation is therefore expected to occur during cold temperature advection and surface evaporation during cold temperature
advection is found to be positive with a mean value and standard deviation of $0.13 \pm 0.06 \, \mathrm{mm \, h^{-1}}$ (Fig. 5b), increasing towards
the equator due to the SST-dependence of ocean evaporation. Small amounts of rainfall are associated with cold temperature
advection (mean value of $0.07 \pm 0.03 \, \mathrm{mm \, h^{-1}}$) and are mainly due to shallow convection behind the cold front. The net air-sea
moisture flux during cold temperature advection is from the ocean into the atmosphere (Fig. 5b).

Warm temperature advection frequently occurs in a few areas in the Southern Ocean where warm air is transported over a
relatively colder ocean. Warm temperature advection hot spots of up to 50% occurrence frequency can be observed north of
the region with highest cyclone frequency and south of the band of high cold temperature advection occurrence frequency
(Fig. 5c). These regions are associated with the warm sectors of extratropical cyclones along the Southern Ocean storm track,
in which warm and moist air is advected polewards. Furthermore, warm temperature advection occurs along the eastern coast
of South America and at 150ºW in the South Pacific, which are regions of anomalously cold ocean waters (Fig. 6). The iso-
lated hot spot in the Pacific is connected to the location of the oceanic polar front, which has its northernmost position between
55 ºS and 60 ºS in the Pacific Ocean around 150 ºW (Moore et al., 1999). The advection of terrestrial and/or subtropical air
over the cold Malvinas current along the Argentinian coast leads to frequent warm temperature advection along the east coast
of South America. During warm temperature advection, surface evaporation is low or negative with a climatological mean of
$0.00 \pm 0.03 \, \mathrm{mm \, h^{-1}}$ (Fig. 5d). Furthermore, warm temperature advection is accompanied by precipitation with a climatological
mean of $0.19 \pm 0.8 \, \mathrm{mm \, h^{-1}}$. Thus, there is a net flux of moisture from the atmosphere into the ocean during warm temperature
advection.



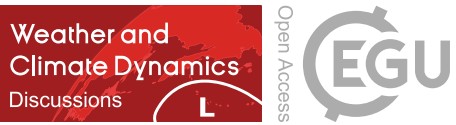

**Figure 5.** Climatological occurrence frequencies of cold and warm temperature advection and zonal flow **(a,c,e)** and the associated ocean evaporation **(b,d,f)** for December to March 1979-2018 using ERA-Interim. Black contours in **(a,c,e)** show climatological cyclone frequencies of 10, 20, 30 and 40 %. Blue dashed lines in **(b,d,f)** show mean surface precipitation in the respective flow category at levels of 0.06, 0.18, 0.24, and 0.3 mm h$^{-1}$ (from light to dark blue, respectively).



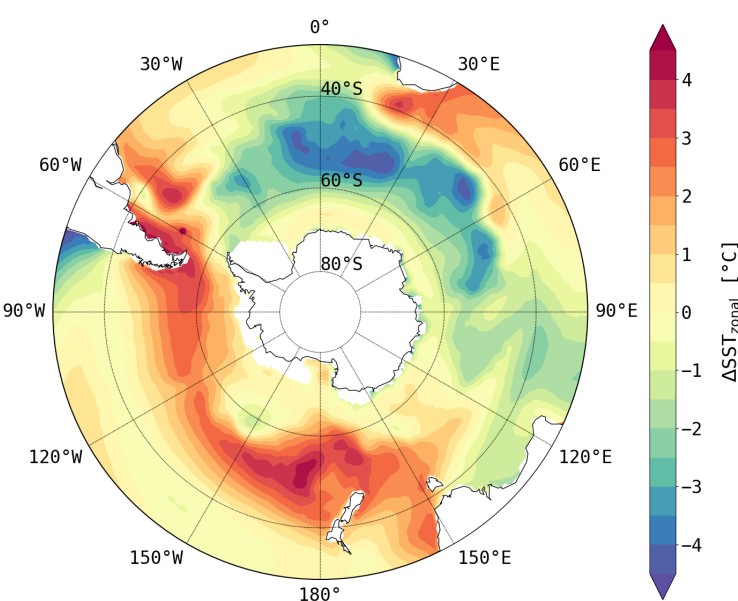

**Figure 6.** Deviations of the climatological sea surface temperature from the zonal mean for the months December to March 1979-2018 using ERA-Interim.

The zonal flow category represents situations when air is advected zonally and ocean surface and air temperature differ by
330 less than $\pm 1°$C. The highest occurrence frequency of zonal advection occurs south of $60°$S (Fig. 5e), where synoptic-scale fronts are rare (Simmonds et al., 2011). Furthermore, zonal flow is frequent at the equatorward edge of the highest cyclone frequency, where air is transported zonally. In the South Pacific, this band of high occurrence frequency between $60°$S and $40°$S spans particularly far north due to the northward shift of the storm track in the Pacific compared to other ocean basins (Wernli and Schwierz, 2006). The composite mean ocean evaporation is low during zonal advection and has a mean value of
335 $0.06\pm0.03$ mm h$^{-1}$, i.e., between the mean values of ocean evaporation during cold and warm temperature advection (Fig. 5f). The mean value of surface precipitation during zonal flow is $0.10\pm0.03$ mm h$^{-1}$ and, therefore, the net moisture flux is slightly negative during zonal flow. Because the net moisture flux during zonal flow is close to zero, we will mostly focus on cold and warm temperature advection in the following.

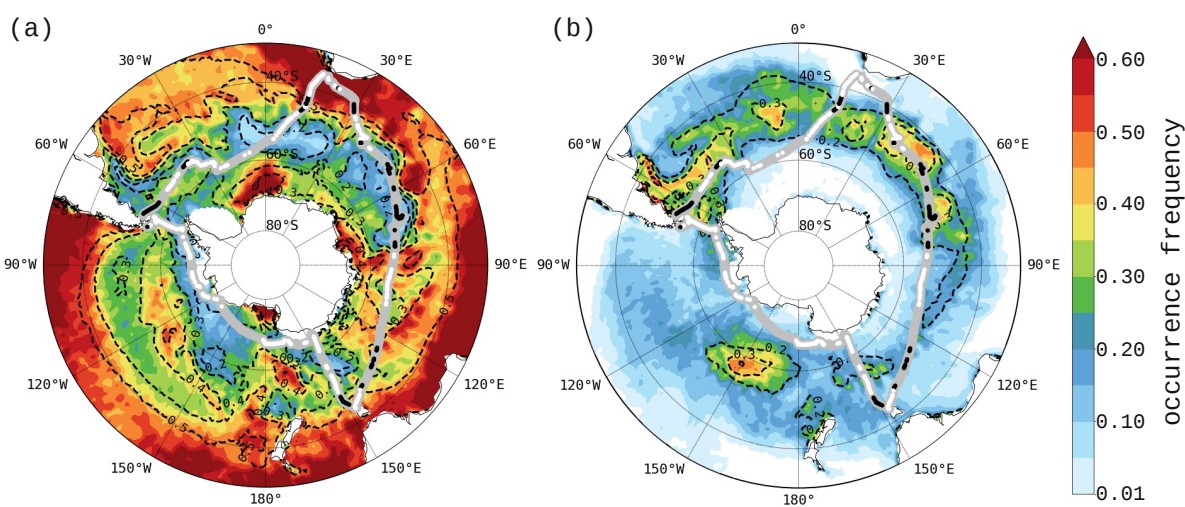

**Figure 7.** Mean occurrence frequency of cold **(a)** and warm **(b)** temperature advection for December 2016 to March 2017 using ERA-Interim. Black, dashed contours show the climatological occurrence frequency of cold **(a)** and warm **(b)** temperature advection as shown in Fig. 5a,c, respectively. The grey thick line shows the ACE ship track with the observed occurrences of cold (white points) and warm (black points) temperature advection events during ACE.

## b) Flow regimes along the ACE track

The spatial patterns of cold temperature advection, warm temperature advection, and zonal flow south of $30\,^{\circ}$S in the ACE summer (Dec 2016 - March 2017, Fig. 7) are in general agreement with the December - March climatology over the period 1979-2018 (Fig. 5a,c). Notable differences include more frequent cold temperature advection events, and especially many cold temperature advection events across Queen Maud Land near $0^{\circ}$ E, during the ACE period. Overall, the frequencies of cold and warm temperature advection events in the entire Southern Ocean during ACE are representative for Southern Hemisphere summer conditions.

 However, specifically along the ACE track, the occurrence frequencies of the three flow regimes differ from the climatology. In the ACE measurements, 59% of all advection events were zonal, 27% cold and 14% warm temperature advection events (see black and white dots in Fig. 7a,b). The frequency of cold temperature advection events along the ACE track is nearly 10% lower than in the climatology for the region south of $30\,^{\circ}$S (Fig. 7a). The ACE track was close to Antarctica only in the Pacific, which means that cold air outbreaks in the Atlantic and Indian Ocean, where the ACE track was mostly located in areas with zonal and warm temperature advection, are undersampled (see also Fig. 7). Warm temperature advection events were mainly encountered in the South Indian and Atlantic Ocean. Therefore, insight from the ACE data set on warm temperature advection is representative for these two ocean basins around Antarctica. Although zonal flow events dominated, a total measurement time of 462 h during cold and 238 h during warm temperature advection, respectively, is available, which provides an observational

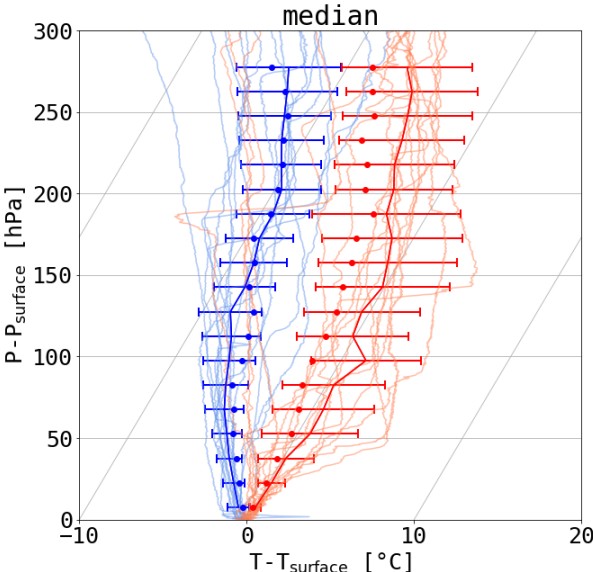

**Figure 8.** Skew-$T$-log-$p$-diagram of air temperature relative to the surface air temperature from radiosoundings (thin lines) during cold (blue) and warm (red) temperature advection events. The y-axis shows the pressure relative to the surface pressure. The thick lines and error bars show the median and standard deviation in every 15 hPa pressure bin. The dots show the median profiles from COSMO$_{iso}$ for the same cold and warm temperature advection events as sampled by the radiosoundings.

data set that is large enough to statistically analyse the typical isotope signature associated with these events.

### c) Vertical temperature profiles and precipitation

The strongly differing environmental conditions during cold and warm temperature advection, as seen in the ERA-Interim
composite analysis (Fig. 5), can also be observed in the ACE measurements. A distinctively different $h_s$ during cold compared to warm temperature advection was observed (see e.g. Fig. 10d) with a median value of 70.9% during cold and 96.9% during warm temperature advection, respectively. This is in agreement with the results from the ERA-Interim composites, which show the strongest positive air-sea moisture fluxes in the cold temperature advection regime and low or negative fluxes in the warm temperature advection regime.
The vertical temperature profiles also differ strongly between cold and warm temperature advection. Radiosoundings during cold temperature advection show a conditionally unstable MBL up to 130 hPa a.s.l. (Fig. 8). With a median surface pressure of 988 hPa, this corresponds to a median MBL top at 848 hPa. During warm temperature advection, the median profile from radiosoundings shows a stable MBL starting from the surface. The individual soundings during warm temperature advection are very diverse. Most of them show a strong temperature inversion below 50 hPa a.s.l., which corresponds to a MBL top at
$930-960$ hPa, indicating a shallow MBL. The entire MBL is, thus, influenced by the opposite air-sea heat fluxes and resulting





mixing processes during cold and warm temperature advection.

As indicated by the ERA-Interim climatology (Fig. 5b,d), precipitation characteristics differ between cold and warm temperature advection. The precipitation rates along the ACE track (derived from MRR measurements) show that during cold temperature advection, rainfall ($> 0\,\mathrm{mm\,h^{-1}}$) occurred during 25% and snowfall during 14% of the time. The median value of the rainfall rate was $0.16\,\mathrm{mm\,h^{-1}}$ and for the snowfall rate $0.04\,\mathrm{mm\,h^{-1}}$. During warm temperature advection, rainfall was present 32% of the time and no snowfall occurred, leading to a smaller total precipitation occurrence frequency than during cold temperature advection. However, precipitation during warm temperature advection was more intense with a median value of $0.31\,\mathrm{mm\,h^{-1}}$. These observations agree with the rainfall intensities observed in the ERA-Interim composites (Fig. 5b,d) with typically heavier precipitation during warm than cold temperature advection. Even though precipitation during cold temperature advection is less intense, it occurs more often resulting in a larger input of precipitation into the ocean. The larger precipitation totals during cold compared to warm temperature advection events is mainly due to the difference in the average geographical extent of the two advection regimes. Cold temperature advection generally occurs over much larger areas than warm temperature advection, which is usually confined to the warm sectors of extratropical cyclones which leads to shorter sections of warm temperature advection along the ship track, respectively (black dots in Fig. 7b).

Overall, the observed environmental conditions during ACE are in agreement with the climatological composite analysis of cold and warm temperature advection based on ERA-Interim. In the next Sections, we will discuss the isotopic signature during cold temperature advection, warm temperature advection and zonal flow (Sect. 4.2) and the processes shaping the differing isotopic signature of MBL water vapour during cold and warm temperature advection (Sect. 4.3).

### 4.2 Observed and simulated SWI composition during the different advection regimes

#### a) ACE observations

To identify the characteristic SWI signal during cold and warm temperature advection, the measured isotope and environmental variables during ACE are analysed with respect to the different temperature advection regimes. The distributions of $\delta^2$H, $\delta^{18}$O and $d$ during warm temperature advection, cold temperature advection and zonal flow are shown in Fig. 9. For all isotope variables, the distributions associated with the cold and the warm temperature advection regime are significantly different when applying a Wilcoxon rank-sum test ($p <0.01$). The mode of the $d$ distribution during cold temperature advection is $10\,‰$ higher than during warm temperature advection. The median $d$ during cold temperature advection is $6.5\,‰$ compared to $-0.4\,‰$ during warm temperature advection. The distribution of $\delta^2$H and $\delta^{18}$O are similar with median values $23.2‰$ and $3.6‰$ higher during warm compared to cold temperature advection, respectively. The mode of the distributions is higher by $28\,‰$ for $\delta^2$H and $4\,‰$ for $\delta^{18}$O during warm than during cold temperature advection. The median and mode of the zonal flow lie in between the respective values of the cold and warm temperature advection distributions for all isotope variables.

As already shown in previous studies (Uemura et al., 2008; Pfahl and Wernli, 2008; Steen-Larsen et al., 2014b; Benetti et al., 2015; Bonne et al., 2019; Thurnherr et al., 2020), $d$ in the MBL is anti-correlated with the near-surface relative humidity. In the ACE measurements, $d$ and $h_s$ negatively correlate with a Pearson correlation of -0.73. During cold temperature advec-



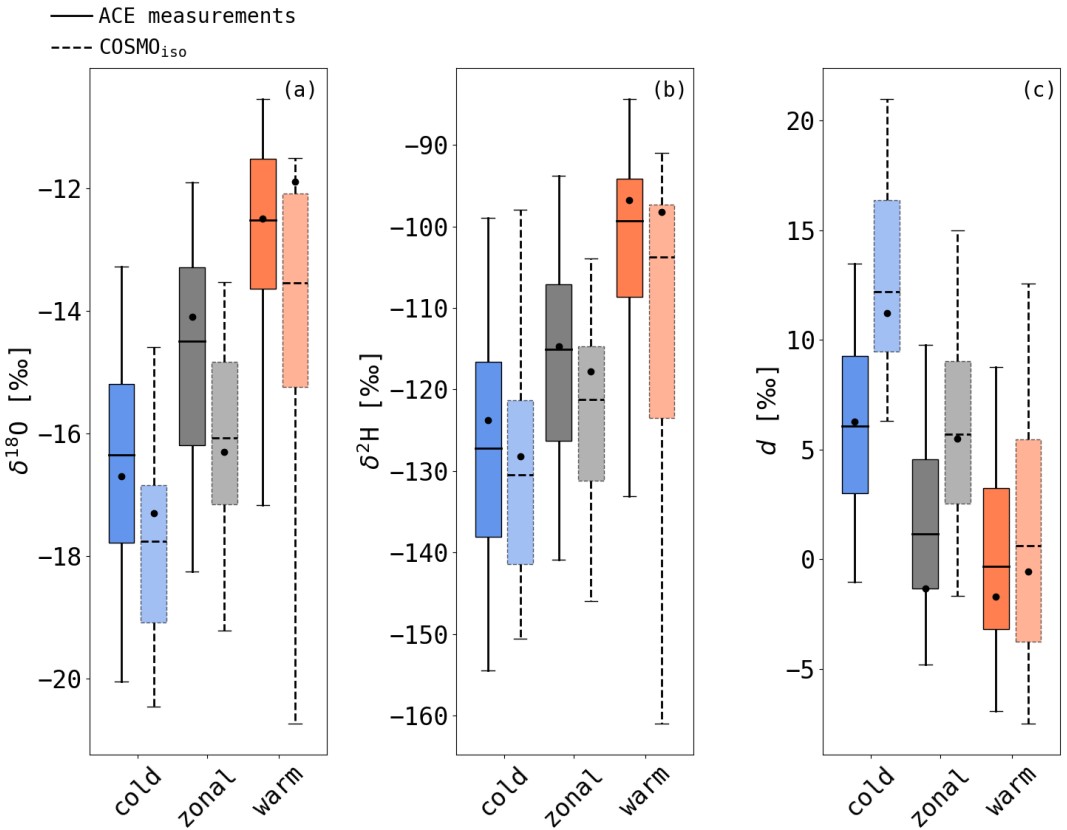

**Figure 9.** Box plots showing median (black horizontal line in box), interquartile range (boxes), mode (black dots) and [5,95]-percentile range (whiskers) of **(a)** $\delta^{18}$O, **(b)** $\delta^2$H and **(c)** $d$ from ACE measurements (solid lines, dark colours) and COSMO$_{\mathrm{iso}}$ simulations at the lowest model level (dashed lines, light colours) during cold temperature advection (blue), zonal flow (grey) and warm temperature advection (orange).

tion, the atmosphere is undersaturated (low $h_s$), and during warm temperature advection close to saturation or oversaturated

($h_s \geq 100\%$, Fig. 10d). Therefore, contrasting atmosphere-ocean moisture fluxes, which can even be of opposite sign, can be associated with cold and warm temperature advection. The ACE measurements confirm the expected contrasts in the isotopic signature and the close link of $d$ and $h_s$ also in oversaturated conditions.

**b) COSMO$_{\mathrm{iso}}$ simulations**

From the air-sea fluxes associated with the different temperature advection regimes, we expect the MBL to be strongly influenced by ocean evaporation during cold temperature advection whereas dew deposition on the ocean surface plays a major role in shaping the observed isotopic composition of water vapour during warm temperature advection. To better understand how





the observed anomalies in the isotope signals form during cold and warm temperature advection, the isotopic composition as well as other environmental variables are analysed along the ACE track using COSMO$_{iso}$ simulations. For 1% of all 1-hourly measurement points of legs 1-3, the classification of cold and warm temperature advection according to the COSMO$_{iso}$ simulations disagrees with the observed classification (see black crosses in Fig. 10). These measurement points and their associated trajectories are excluded from the following analysis. The very low $\Delta T_{ao} < -5.0°C$ and very high $\Delta T_{ao} > 7.0°C$ in the ACE measurements are not seen in COSMO$_{iso}$ (Fig. 10e). These instances belong to a katabatic wind event and a vertical dry intrusion event for which COSMO$_{iso}$ did not correctly simulate the meteorology.

The simulated isotope variability is in agreement with the measurements with a Pearson correlation coefficient $\rho$ of 0.74 for $\delta^2$H, 0.68 for $\delta^{18}$O and 0.68 for $d$ (Fig. 10). COSMO$_{iso}$ reasonably reproduces the measured SWI variability and composition during ACE. The same qualitative distribution of cold and warm temperature advection SWI signal is seen with a shift in $\delta^2$H and $\delta^{18}$O towards negative values for all temperature advection regimes and a shift in $d$ towards positive values during cold temperature advection and zonal flow compared to the measured composition during ACE (Fig. 9). This difference specifically during conditions with important contributions of water vapour to the MBL by ocean evaporation could be caused by too strong vertical mixing in the COSMO$_{iso}$ simulations. This effect could also lead to the slightly lower specific and relative humidity in the simulation compared to the measurements (Fig. 10d,f). The vertical temperature gradient in the MBL can be used as a measure of vertical mixing. During cold advection, the vertical temperature structure in the MBL is generally simulated well by COSMO$_{iso}$ (Fig. 8). The simulated median vertical temperature profile shows a temperature inversion around 150 hPa a.s.l., which lies above the inversion in the measured profiles at 130 hPa a.s.l. This supports the hypothesis that COSMO$_{iso}$ has too strong vertical mixing as a higher inversion height implies more mixing. Furthermore, too strong entrainment at the MBL in COSMO$_{iso}$ could also contribute to the observed difference in SWIs during cold temperature advection. These findings show that the environmental conditions during cold temperature advection, i.e. during conditions of strong ocean evaporation, are not well reproduced in COSMO$_{iso}$. The simulated median temperature profile during warm advection shows, in accordance with the measured profile, a stable MBL. Due to the model's vertical resolution, very strong temperature inversion in the radiosoundings are not represented in the simulated profiles. This is most likely the reason for the negative temperature bias in the simulated profiles during warm advection.

A further reason for the differences between the measurements and simulations could originate from the formulation of non-equilibrium isotopic fractionation in COSMO$_{iso}$. Using a weaker, wind-independent formulation of the non-equilibrium fractionation factor by Merlivat and Jouzel (1979) (in the smooth regime at $6\,\mathrm{m\,s^{-1}}$) instead of the currently used formulation by Pfahl and Wernli (2009) in COSMO$_{iso}$ simulations with parametrised convection leads to a decrease of $d$ by, on average, 2‰, on the lowest model level over the ocean surface (Jansing, 2019). The larger negative bias in $\delta^{18}$O compared to $\delta^2$H in COSMO$_{iso}$ could also be explained by too strong non-equilibrium fractionation using the formulation by Pfahl and Wernli (2009). However, the simulations using the formulation by Merlivat and Jouzel (1979) also show a decrease in $d$ variability above oceanic areas (Jansing, 2019). As we are interested in the processes shaping the SWI variability in the MBL, the formulation by Pfahl and Wernli (2009) is more adequate to use here as the SWI variability in the measurements and simulations agrees well.





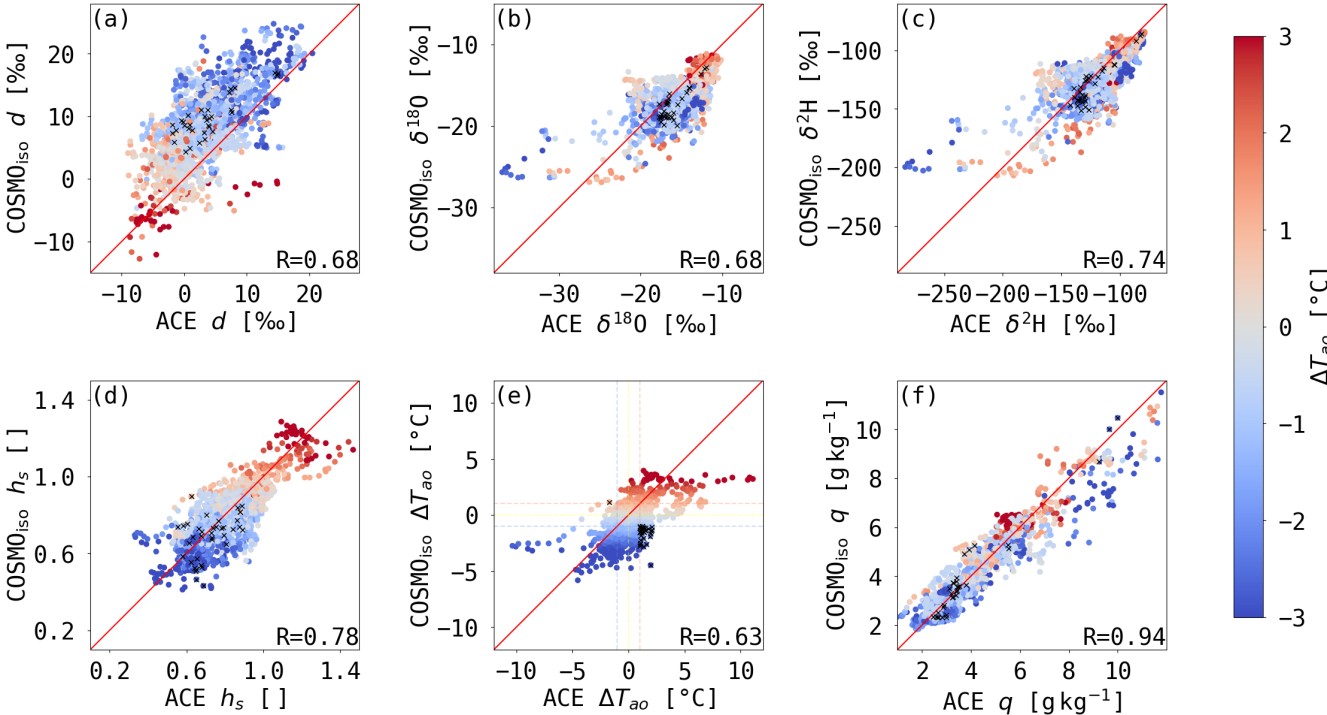

**Figure 10.** Scatterplots of 1-hourly ACE measurements vs. interpolated values from COSMO$_{\text{iso}}$ simulations at the lowest model level along the ACE ship track of **(a)** $d$, **(b)** $\delta^{18}$O, **(c)** $\delta^{2}$H, **(d)** $h_s$, **(e)** air-sea temperature difference $\Delta T_{ao}$, and **(f)** specific humidity $q$ coloured by $\Delta T_{ao}$ from COSMO$_{\text{iso}}$. Points with contradicting cold and warm temperature advection classification between the measurements and simulations are marked with black crosses. The R value of the Pearson correlation for each variable is shown in the bottom right of the panels.

To better understand the difference in SWIs between measurements and COSMO$_{\text{iso}}$ simulations, further studies are needed such as the detailed analysis of case studies based on extensive 3D-data sets as, for example, collected during the Iceland

Greenland Seas Project (Renfrew et al., 2019). Even though near-surface simulated and measured isotope signals do not agree everywhere, the COSMO$_{\text{iso}}$ simulations capture the observed variability of the isotopic composition and provide similar distributions of isotope variables as the observations for the three advection categories. These simulations will thus be used for an assessment of the relevant processes shaping the isotopic composition of water vapour in the MBL during cold and warm temperature advection.

455

### 4.3   Formation of isotope anomalies during cold and warm temperature advection

During transport, the specific humidity of air masses varies due to different moist atmospheric processes such as ocean evaporation, dew deposition, cloud formation, and below-cloud evaporation. These processes might alter the isotopic composition of the water vapour substantially between the moisture source and the point of measurement. With a Lagrangian composite anal-





ysis of cold and warm temperature advection events, we aim to assess the relative importance of these processes in shaping the SWI composition of water vapour in the MBL. One way to analyse how such moist processes during transport affect the SWI composition of the air mass is to compare the air mass' properties at the moisture source and upon arrival at the measurement site (Fig. 11). The weighted mean of the moisture source properties of the MBL water vapour along the ACE track is computed using 7-day backward trajectories from 3D COSMO$_{\text{iso}}$ wind fields and is compared to the properties of the trajectories at the arrival and at the driest point along the back-trajectories.

### a) Cold temperature advection

For cold temperature advection, the moisture uptake took place 36 [45,25] h (the numbers in the brackets denote the [25%,75%] percentile range) before the air parcels arrived at the measurement site (Fig. 11e). The air parcels enter the cold sector 50 [91,31] h before arrival and, thus, the bulk of the moisture of the cold air is taken up in or shortly before entering the cold sector. After the uptake, small changes in $\delta^2$H, $\delta^{18}$O and $d$ between the moisture source and the arrival are seen (Fig. 11a), suggesting that other post-evaporation processes such as cloud formation or interaction with precipitation have a limited impact on the observed isotope composition at the ship location. $\delta^2$H, and to a smaller extent $\delta^{18}$O, show a weak increase from the moisture source until arrival (Fig. 11b,c), while the air moves equatorwards (Fig. 11d and the cold temperature advection trajectories in Fig. 4). This isotopic enrichment might be due to the weaker equilibrium fractionation at higher SST closer to the arrival, which leads to higher $\delta^{18}$O and $\delta^2$H in the evaporated water vapour. A much stronger increase in $\delta^2$H and $\delta^{18}$O is seen between the minimum $q$ along the 7-day backward trajectories, which has a median value of 1.3 g kg$^{-1}$, and the moisture source, where $q$ has a median value of 2.7 g kg$^{-1}$, showing how strongly the advected SWI signal in the MBL water vapour is changed by the moisture uptake. A similar median $d$ can be observed at the minimum $q$, the moisture source location and at the ship's position, while the moisture source location shows a wider interquartile range than the other two locations along the trajectories. The higher variability in $d$ at the moisture source compared to the location of minimum $q$ could be caused by the large variability in ocean evaporation and air temperature at the moisture source locations for water vapour arriving along the ACE track (not shown). Furthermore, $d$ at the moisture source is the median of the weighted mean conditions of all moisture uptakes, which can spread over a large region. This leads to a wider distribution of $d$ at the moisture source then for $d$ along the ACE track. Overall, the isotopic composition of water vapour in the cold sector is strongly affected by the moisture uptake within the sector, which overwrites the advected SWI signal.

### b) Warm temperature advection

For warm temperature advection, the moisture uptake occurs 49 [40,61] h before arrival, while the air parcels enter the warm sector much later at 20 [11,30] h before arrival (Fig. 11e). Therefore, the air parcels take up moisture upstream of the warm sector of an extratropical cyclone, generally in a region with cold temperature advection and sometimes in a region of zonal flow. Similar to cold temperature advection, the isotopic composition at minimum $q$ along the 7-day backward trajectories of a





**Figure 11.** Box plots with information derived from COSMO$_{\mathrm{iso}}$ trajectories for cold (blue) and warm (orange) temperature advection events. Boxes show median (black horizontal line in box), interquartile range (boxes) and [5,95]-percentile range (whiskers) of **(a)** $d$, **(b)** $\delta^{18}$O, **(c)** $\delta^2$H and **(d)** latitude $\phi$ at minimum specific humidity along the trajectories (minq), the moisture source site (ms) and ship location (ship), **(e)** the weighted mean moisture uptake time before arrival (uptake) and the entrance time into the cold and warm sectors (sector), and **(f)** the specific humidity fraction explained by the moisture source attribution.





median value of $3.1 \, \mathrm{g \, kg^{-1}}$ is strongly altered due to ocean evaporation at the moisture source, where a median $q$ of $5.7 \, \mathrm{g \, kg^{-1}}$ is seen. In contrast to the cold sector, not only $\delta^{18}$O and $\delta^2$H, but also $d$ increases from minimum $q$ to the moisture source

location. This increase in median $d$ could be caused by a stronger median increase in latitude between minimum $q$ and the moisture source during warm temperature advection compared to cold temperature advection (Fig. 11d). Median $d$ changes by $-10‰$ from the moisture source to arrival, revealing that the isotopic composition of the water vapour can be strongly modified in the warm sector, for example due to cloud formation, precipitation or dew deposition. The strength in $d$-decrease between the moisture source and measurement location depends on the residence time in the warm sector. There is a weak trend

($\rho$=0.31) towards a stronger decrease in $d$ with an increase in residence time in the warm sector (not shown). For a residence time of, for example, $40 \, \mathrm{h}$ in the warm sector, there is a decrease in $d$ of $11‰$ between the moisture source and the arrival at the measurement site. During warm temperature advection, the air moves poleward from the moisture source (Fig. 11d and warm temperature advection trajectories in Fig. 4) and shows a weak increase in $\delta^{18}$O from the moisture source to the point of measurement along the ship track. $\delta^2$H stays at a similar median value, but shows a wider distribution upon arrival compared to

the moisture source. To better understand these changes between the isotopic composition at the moisture source and the point of measurement, the temporal evolution of the isotopic composition and other environmental variables are analysed along the backward trajectories for the 4 days before arrival.

### c) Temporal evolution of SWI signals along cold advection trajectories

The temporal evolution of $d$, $\delta^{18}$O and $\delta^2$H along the backward trajectories in the cold sector shows a continuous increase (Fig. 12a-c). These changes occur simultaneously with an increase in ocean evaporation, an equatorward movement and descent of the air (Fig. 12d-f). The strongest changes in $d$ can be observed in the cold sector during the last 48 hours before arrival, when the air masses are closest to the ocean surface and the ocean evaporation and specific humidity increase strongly (Fig. 12d,g). The changes of the isotopic composition along the trajectories can be described by two stages. During the first stage, only

$\delta^2$H and $\delta^{18}$O increase and $d$ stays constant. In this period until approximately $40 \, \mathrm{h}$ before arrival, the increase in the $\delta$-values might be mainly caused by mixing during the descent with air masses at lower altitudes with higher $\delta$ values and a similar $d$ as the descending air masses, which does not affect $d$ but leads to an increase in $\delta^{18}$O and $\delta^2$H. The second stage starts once the trajectories are closer to the sea surface and within the cold sector, where $E$ and $d$ increase strongly. During this second stage, the air masses are more strongly influence by ocean evaporation and, thus, non-equilibrium fractionation, which leads

to an increase in $\delta^{18}$O and $\delta^2$H as well as $d$. The air masses arriving in the cold sector are weakly influenced by cloud and precipitation-related processes (Fig. 12h,i and compare dashed and dotted blue lines in Fig. 12a-i). Precipitating air parcels arriving during cold temperature advection, i.e. showing surface precipitation $>0.1 \, \mathrm{mm \, h^{-1}}$ upon arrival, have a lower $d$ and a small decrease in pressure upon arrival compared to the median values for all cold temperature advection trajectories. This small change in $d$ cannot be explained by changes in $E$ as median $E$ for precipitating trajectories is similar to the median $E$

over all trajectories. During rainfall, which occurs during 68% of all precipitation events, the equilibration of rain droplets with the surrounding water vapour or the (partial) evaporation of rain droplets can lead to the changes in $d$ in water vapour. For



**Figure 12.** Composites of COSMO$_{iso}$ trajectories showing time series along trajectories from cold (blue) and warm (red) temperature advection events, respectively, of median values of **(a)** $d$, **(b)** $\delta^{18}$O, **(c)** $\delta^2$H, **(d)** surface evaporation ($E$), **(e)** latitude ($\phi$), **(f)** pressure ($p$), **(g)** specific humidity ($q$), **(h)** the sum of cloud and ice water content ($q_c+q_i$), and **(i)** the sum of rain and snow water content ($q_r+q_s$). Cold and warm temperature advection events are only considered over open ocean (no sea ice, land fraction <0.1 in COSMO$_{iso}$). Shadings denotes the [25,75] percentile range and vertical lines show the median time step when the trajectories enter the cold (blue line) and the warm sector (red line). Furthermore, median values are shown for trajectories experiencing no surface precipitation upon arrival (total surface precipitation $R_{tot} < 0.01\,mm\,h^{-1}$; dashed lines) and for those with surface precipitation upon arrival ($R_{tot} > 0.01\,mm\,h^{-1}$; dotted lines).





cold temperature advection events without precipitation, there is a slightly higher $d$, which could be caused by the absence of interaction between rain and water vapour. Thus, the signal from ocean evaporation is more dominant for the non-precipitating trajectories. Overall, the difference between the precipitating and non-precipitating trajectories in the cold sector is small and

lies within the standard deviation for all variables. Therefore, the positive anomalies in $d$ due to ocean evaporation in the cold sector are not substantially altered by precipitation-induced changes in the isotopic composition of the MBL water vapour.

The small difference between precipitating and non-precipitating cold temperature advection events can also be observed in the vertical SWI profiles along the ACE ship track, which show a lower $d$ and higher $\delta$ values during precipitating than non precipitating events throughout the MBL (Fig. 13a-c). The equivalent potential temperature ($\Theta_e$) within the MBL indicates

well-mixed conditions (Fig. 13d) and a strong influence of ocean evaporation on the MBL moisture budget. This is reflected in the SWI profiles which show constant values in $d$, $\delta^{18}$O and $\delta^2$H in the lower MBL. In the upper MBL, $\delta^{18}$O and $\delta^2$H show a weak decrease, which progresses further above the MBL height, implicating vertical mixing of free tropospheric air into the MBL. Even though ocean evaporation is the main process affecting the MBL isotopic composition, cloud processes might affect the region around the MBL top, where $d$ has its minimum values. This minimum occurs above the region of

highest cloud and ice water content ($q_c + q_i$, Fig. 13e) and rain and snow water content ($q_r + q_s$, Fig. 13f). Furthermore, the $d$ minimum is at higher altitudes for precipitating air masses where also the maximum in $q_c + q_i$ is observed at higher altitudes. A minimum of $d$ at the MBL top has been observed in measurements of SWIs and several processes where discussed such as evaporation of cloud and rain droplets (Sodemann et al., 2017; Salmon et al., 2019). A further process which could induce low $d$ in water vapour close to the MBL top is cloud formation during decreasing temperatures such as for example during a moist

adiabatic ascent. The condensation of water vapour in an environment with decreasing air temperature leads to a decrease in $d$ in the remaining water vapour due to the temperature dependency of equilibrium fractionation (see also Appendix A). More detailed studies of these processes comparing measurements, e.g. on board aircrafts, and model simulations are needed to better understand the processes involved in these low $d$ values at the MBL top.

**d) Temporal evolution of SWI signals along warm advection trajectories**

For warm temperature advection, $\delta$-values increase along the trajectories before the air masses arrive in the warm sector. Within the warm sector, in the last 20 hours before arrival, the $\delta$-values start to decrease, with $\delta^2$H starting earlier than $\delta^{18}$O (Fig. 12b,c). The $d$ already starts decreasing around 60 h before arrival. During the decrease in $d$ outside of the warm sector, the air masses descend and ocean evaporation decreases while there is only a small increase in $q$ (Fig. 12f,g). Furthermore, the

movement of the air masses changes from equatorward to poleward (Fig. 12e). Therefore, this episode of decreasing $d$ outside of the warm sector could be due to weaker non-equilibrium fractionation during ocean evaporation. During the $d$-decrease within the warm sector, the air parcels stay at the same altitude or ascend slightly, while $E$ is close to zero or changes sign implying dew formation. The median precipitation during warm temperature advection is close to zero but can occasionally exceed $50\,\mathrm{mg\,kg^{-1}}$ and shows rainfall for 98% of all precipitation events. Warm temperature advection trajectories experi-

encing precipitation upon arrival show a stronger decrease in $d$, $\delta^2$H and $\delta^{18}$O in the warm sector than the median values



**Figure 13.** Composites of vertical profiles from COSMO$_{\text{iso}}$ showing the median of **(a)** $d$, **(b)** $\delta^{18}$O, **(c)** $\delta^2$H, **(d)** $\theta_e$, **(e)** the sum of cloud and ice water content ($q_c + q_i$), and **(f)** the sum of rain and snow water content ($q_r + q_s$) for cold (blue) and warm (red) temperature advection events along the ACE ship track. The shading denotes the interquartile range. Furthermore, the median vertical profiles for conditions with (dotted lines) and without (dashed lines) surface precipitation are shown for warm and cold temperature advection. On the y-axis the height relative to the boundary layer height is shown, where 1 denotes the height of the boundary layer (black dashed line).



for all warm temperature advection trajectories. This stronger decrease in SWIs is accompanied by stronger dew deposition, higher values of cloud and rain water content and a tendency for ascent. The evolution of $d$ and the $\delta$-values shows that two different processes can lead to the observed decrease in $d$ in the warm sector. After the entrance into the warm sector, $d$ and $E$ decrease, while $\delta^{18}$O and $\delta^2$H increase. This stage is dominated by dew deposition and non-equilibrium processes during dew

deposition might lead to the observed changes in SWIs. Shortly before arrival, $\delta^{18}$O and $\delta^2$H also start decreasing. While dew deposition is still high, the cloud and rain water content increases strongly before arrival. At this stage, dew deposition as well as the interaction of water vapour with cloud and rain droplets affect the isotopic composition of the water vapour. $d$ in cloud droplets is low during the precipitation events in the warm sector (not shown). Therefore, an exchange between cloud droplets and the surrounding water vapour could lead to a decrease in $d$. Nonetheless, the vertical profiles during warm temperature

advection along the ACE track do not show lower $d$ in regions of high $q_c + q_i$ (Fig. 13a,e). The median vertical $d$ profile as well as the profile for precipitating and non-precipitating events show lowest $d$ close to the surface implying that surface-related processes such as dew deposition are most important in forming negative $d$ anomalies in the warm sector. To better understand the changes in SWIs in the warm sector, the temporal evolution of SWIs in water vapour during the moist processes in warm sector needs to be described using a mechanistic physical approach. A study is in preparation describing the temporal evolution

of the isotopic composition of water vapour with single physical process models.

Even though dew deposition is the most important process in the lower MBL leading to the decrease in $d$ during warm temperature advection, further processes might be important in the upper MBL. This is indicated by the vertical $d$ profile of non-precipitation trajectories. These trajectories show a higher $d$ and weaker dew deposition upon arrival than the median of all trajectories (Fig. 12a,d), but still show the typical decrease in $d$ in the warm sector due to low ocean evaporation or dew

deposition. Furthermore, a specific difference can be observed for precipitating and non-precipitating air masses in the vertical profiles. The non-precipitating profiles show a minima of $d$ close to the surface and a stagnation in $d$-increase with height in the upper MBL (Fig. 13a). The minimum close to the surface is most likely related to dew deposition similar to the precipitating trajectories. The stagnation in the upper MBL does not correspond to a region of enhanced precipitation or cloud occurrences upon arrival. Therefore, this weaker increase in $d$ in the upper MBL might have been caused by upstream processes as for

example the moist adiabatic ascent of the air parcel and subsequent cloud formation similar to the $d$ minimum at the MBL top in the cold sector. Due to the diverse Lagrangian history of the different warm temperature advection events, detailed case studies are needed to study such upstream processes and to understand how important the advection of low $d$ signals in the upper MBL are. In summary, the MBL water vapour during warm temperature advection is strongly affected by dew deposition in the warm sector, which mainly influences near-surface air masses due to the stable MBL (see Fig. 13d). Cloud processes can

influence $d$ in water vapour in the upper MBL, and the SWI signal might be advected from upstream processes.

The comparison of the isotopic composition during cold and warm temperature advection in COSMO$_{\mathrm{iso}}$ simulations shows that ocean evaporation and dew deposition are the main processes affecting SWIs in water vapour in the MBL. The impact of local precipitation on the SWIs in the MBL is small compared to the anomalies induced by air-sea interactions. Still, negative $d$ anomalies can be seen in the upper MBL and might be caused by cloud and precipitation-related processes. The importance



of such processes, such as cloud formation and below-cloud processes, needs more detailed future studies which include the comparison of measurements and model data in the upper MBL.





## 5 Discussion and Conclusions

The aim of this study was to systematically quantify the contrasting isotopic composition in MBL water vapour in the cold and warm sectors of extratropical cyclones in the Southern Ocean, and to identify the main processes that are responsible for these signals. In order to address these objectives in a robust way, i.e., by averaging over many cold and warm sectors, the following prerequisites were indispensable: (i) a method to objectively identify cold and warm sectors of extratropical cyclones that can be applied to ship measurements as well as to reanalysis data (here we used a simple approach focusing on air-sea temperature differences due to cold and warm temperature advection, respectively; Hartmuth, 2019); (ii) an extended data set of SWI measurements available from the three-month Antarctic Circumnavigation Experiment in 2016/2017 (Thurnherr et al., 2020), which contains observations from 29 cold and 18 warm advection events corresponding to 462 h and 238 h of data, respectively; and (iii) simulations with the high-resolution, SWI-enabled numerical weather prediction model COSMO$_{\text{iso}}$ which allows the calculation of air parcel trajectories and studying the processes affecting SWI signals along the flow (Pfahl et al., 2012). The simulated SWI signals agree well with the ship-based measurements, in particular, in terms of the observed variability in $\delta^{18}$O, $\delta^2$H and $d$ at the synoptic time scale, enabling the joint analysis of observations and simulations. Only the combination of these diagnostic, observational and modelling elements made it possible to provide a portrayal of the characteristic SWI signals associated with warm and cold advection events and their underlying physical processes.

The analyses in this study show that the cold and warm sectors of extratropical cyclones are associated with contrasting isotopic signals in MBL water vapour. The main conclusions can be summarized as follows, separately for situations with cold and warm temperature advection, i.e. for cold and warm sectors of extratropical cyclones, respectively.

- In the cold sector, negative $\delta^{18}$O- and $\delta^2$H-anomalies (i.e. deviations from the campaign mean) and positive $d$-anomalies occurred together with low $h_s$ and a deep and unstable MBL. The trajectory analysis based on COSMO$_{\text{iso}}$ simulations showed that during cold temperature advection, the moisture uptake due to ocean evaporation occurs typically $25-45$ h before arrival of the considered air parcels at the measurement site. This moisture originates from ocean evaporation mainly within the cold sector itself, and ocean evaporation is the main process that shapes the isotopic composition of the measured water vapour. Moreover, ocean evaporation in the cold sector influences the SWI composition of the entire MBL leading to an increased vertical gradient of $\delta^2$H and $\delta^{18}$O between the MBL and the free troposphere.

- In the warm sector, positive $\delta^{18}$O- and $\delta^2$H-anomalies and negative $d$-anomalies were observed during meteorological conditions with high $h_s$ and a shallow and stable MBL. Processes shaping the SWI composition during warm temperature advection as identified using COSMO$_{\text{iso}}$ simulations are more diverse than during cold temperature advection. The air parcels enter the warm sector typically $11-30$ h before arrival at the measurement site, which generally occurs after the moisture uptake that is most prominent in the time window $40-61$ h before arrival. Therefore, the uptake of moisture that ends in the warm sector happens outside of the warm sector in a region of cold temperature advection and affects the isotopic composition of the air parcels with an increase in $\delta^2$H, $\delta^{18}$O and $d$. In addition, within the warm sector, the air parcels encounter a net loss of moisture due to two main processes. First, air-sea interactions in the form of dew





deposition leads to a negative $d$-anomaly and positive $\delta^{18}$O- and $\delta^2$H-anomalies. Second, the formation of clouds and
precipitation in ascending air parcels decreases $d$ in water vapour. Thus, in summary, the characteristic SWI signals
associated with the warm sector are shaped by both, the processes during moisture uptake outside the warm sector, in a
context of cold advection, and subsequent dew and cloud formation within the warm sector.

So far, only few studies have discussed the occurrence of low $d$ in MBL water vapour on synoptic timescales. Kurita et al.
(2016) analysed SWI measurements in MBL water vapour along the East Antarctic coast. They showed similar contrasting
SWI signals over the open ocean as observed in this study with low or negative $d$ and high $\delta^2$H in poleward moving warm
air and high $d$ and low $\delta^2$H in equatorward moving air of Antarctic origin. Furthermore, they showed that the $\delta^2$H variability
along the East Antarctic coast is linked to the southward movement of cyclones, which advect warm and moist air towards the

Antarctic continent, potentially contributing to heavy precipitation (Gorodetskaya et al., 2014; Welker et al., 2014). We showed
here, that these contrasting SWI signals cannot only be found close to Antarctica, but throughout the Southern Ocean and that,
over the open ocean, the synoptic timescale variability of SWI signals is the result of strongly varying air-sea interactions
induced by the meridional advection of air masses within extratropical cyclones. The contrasting air-sea fluxes in cold and
warm sectors of extratropical cyclones have been observed in previous ship measurement campaigns, i.e., during pre-ERICA

(Neiman et al., 1990) and FASTEX (Persson et al., 2005) in the North Atlantic.

Low or negative $d$ in the boundary layer water vapour has also been observed over ice-covered areas due to the deposition
of water vapour on the snow surface (Bonne et al., 2019), where it can impact the isotopic composition of the surface snow
(Steen-Larsen et al., 2014a; Casado et al., 2018; Madsen et al., 2019). The here observed $d$-anomalies in the MBL over the
open ocean due to air-sea interactions during warm and cold temperature advection could also lead to changes in the isotopic

composition of the ocean surface waters or, even more prominently, precipitation formed from these air masses. It is an impor-
tant implication of the results in this study that clouds forming in the cold sector of extratropical cyclones (typically shallow
convective cloud) and clouds forming in the warm sector (typically in the rapidly, slant-wise ascending warm conveyor belts)
rely on different SWI "starting conditions", i.e. on vapour with contrasting SWI anomalies. A recent study (Aemisegger, 2018)
found a clear link between $d$ in monthly precipitation in Reykjavik (Iceland) and the frequency and location of North Atlantic

cyclones, which influences the location of strong ocean evaporation during cold temperature advection. It has yet to be studied
if and how air-sea fluxes in the context of warm temperature advection affect the isotopic composition of precipitation, for ex-
ample, in warm conveyor belts. Future collocated measurements of dew and fog over the ocean during warm advection events
will provide key insights into the coupling between clouds and their environments. Previous studies over land showed that the
time evolution of the isotopic composition of cloud droplets in fog was mainly driven by the moisture origin at large scales

and the enrichment of the fog dependent on the occurrence of precipitation within the cloud (Spiegel et al., 2012a, b). Future
detailed studies of simultaneous cloud water, vapour, precipitation and dew over the ocean within near-surface clouds would
provide important empirical data for the validation of model isotope microphysics.

A previous, highly idealized study based on COSMO$_{\text{iso}}$ simulations without isotopic fractionation during ocean evaporation
(Dütsch et al., 2016) has shown that $\delta^2$H in the cold and warm sectors of extratropical cyclones are primarily affected by

horizontal transport. Here, we confirm that the meridional large-scale transport of air strongly affects the isotopic composition




of the MBL water vapour, based on observations and real-case simulations. In particular, $\delta^{18}$O and $\delta^2$H in water vapour in the cold and warm sectors of extratropical cyclones are influenced by ocean evaporation at the moisture source. The moisture uptake over the ocean surface leads to an increase in $\delta^{18}$O and $\delta^2$H and this SWI signal is only marginally changed during transport to the measurement site. Surprisingly, and in contrast to $\delta^{18}$O and $\delta^2$H, $d$ shows large variations after the moisture

uptake in the warm sector indicating that $d$ in the warm sector is not only influenced by the moisture source conditions, but is changed substantially during transport. Therefore, the isotopic composition of water vapour in the MBL is mainly a signal from air-sea interactions, such as ocean evaporation at the moisture source in the cold sector and dew deposition during transport in the warm sector. The mechanisms leading to isotopic fractionation associated with ocean evaporation, dew deposition and cloud formation as well as the relative contributions of these processes to the observed SWI variability in the cold and

warm sector of extratropical cyclones will be analysed in future studies. Furthermore, the findings of this study are valuable for further studies analysing meridional air mass advection and other characteristics associated with the dynamics of the storm tracks over interannual timescales.

*Data availability.* The ACE data sets are published on the research data repository zenodo: https://zenodo.org/communities/spi-ace/. The

radiosonde data sets is accessible using the following doi: XXX. The rainfall data set is publish with the doi 10.5281/zenodo.3929289 and the snowfall data set with the doi XXX. The COSMO$_{\text{iso}}$ simulation are available on the ETH research collection using the doi XXX.





### Appendix A: Decrease in $d$ due to condensation during a moist adiabatic ascent

The condensation of water vapour in an ascending air parcel leads to the formation of cloud droplets and changes the isotopic composition of the remaining atmospheric water vapour. This process can be modelled using a Rayleigh fractionation model based on the temperature and moisture evolution of a moist-adiabatic ascent of an air parcel. Thereby it is assumed, that the condensed cloud droplets do not interact anymore with the surrounding water vapour after condensation and are immediately removed. This is a simplification of processes occurring during cloud formation, but it provides a first order understanding of isotopic variations due to cloud formation.

A Rayleigh model (Dansgaard, 1964) is used to estimate the effect of condensation during a moist adiabatic ascent on the isotopic composition of water vapour, including the temperature dependency of equilibrium fractionation factor $\alpha_e(T)$ (Horita and Wesolowski, 1994) and the transition from liquid to ice clouds as in Dütsch et al. (2017). An effective isotopic fractionation factor $\alpha_{eff} > 1$ is defined, depending on temperature and the relative fractions of liquid and solid condensate during the moist-adiabatic ascent: $\alpha_{eff} = \alpha_e(T) \cdot f_{liq} + \alpha_{ice} \cdot (1 - f_{liq})$. $f_{liq}$ is 0 for $T < 250.15\,\mathrm{K}$ and 1 for $T > 273.15\,\mathrm{K}$ and a quadratic interpolation is used in between these temperatures to represent mixed phase clouds. $\alpha_{ice} > 1$ is the equilibrium fractionation factor in vapour with respect to ice (Majoube, 1971), adjusted for supersaturation over ice (Jouzel and Merlivat, 1984).

Figure A1 shows the Rayleigh fractionation during a moist adiabatic ascent, which is initiated at a temperature of 283.15 K, a pressure of 1000 hPa and isotopic compositions $\delta^2\mathrm{H}_0 = -90.0\,‰$ and $\delta^{18}\mathrm{O}_0 = -12.0\,‰$ (which corresponds to $d_0 = 6\,‰$). Four different scenarios are calculated:

1. Rayleigh fractionation with constant temperature and a step-wise loss of moisture (blue line).

2. Rayleigh fractionation along a moist adiabatic ascent, i.e. with temperature variations, assuming only liquid clouds (black line).

3. same as 2., including ice cloud formation and the fractionation factor for vapour over ice (green line).

4. same as 3., including an adjustment of the ice fractionation factor due to ice supersaturation (orange line).

All of these four scenarios show a depletion of $\delta^{18}\mathrm{O}$ and $\delta^2\mathrm{H}$ in water vapour with ongoing moisture loss. $d$ in water vapour decreases in the beginning for all scenarios. The decrease in $d$ is small of a value around $1\,‰$ for scenario 1 with constant temperature and increases to values above the start value after the condensation of 75% of the humidity. In the other three scenarios, there is a stronger decrease in $d$ with minimum values of more than $6\,‰$ below the starting value. This stronger decrease in $d$ is due to the temperature-dependency of the equilibrium fractionation factor, which increases more rapidly with decreasing temperature for $^2\mathrm{H}$ than for $^{18}\mathrm{O}$ (see $^2\alpha_{eff}(^{18}\alpha_{eff})^{-1}$ in Fig. A1d). Thus, $d$ in the remaining vapour decreases with decreasing temperature. Scenario 3, which includes the equilibrium fractionation effects of the ice phase but neglects non-equilibrium effects due to supersaturation, shows a weaker decrease in $d$. When including ice supersaturation, the evolution of $d$ follows a similar pattern as scenario 2 with only liquid cloud formation. Once the amount of heavy isotopes in the vapour phase becomes very small, $d$ increases by definition (due to the non-linearity of the $\delta$-scale, see also Dütsch et al., 2017) for all





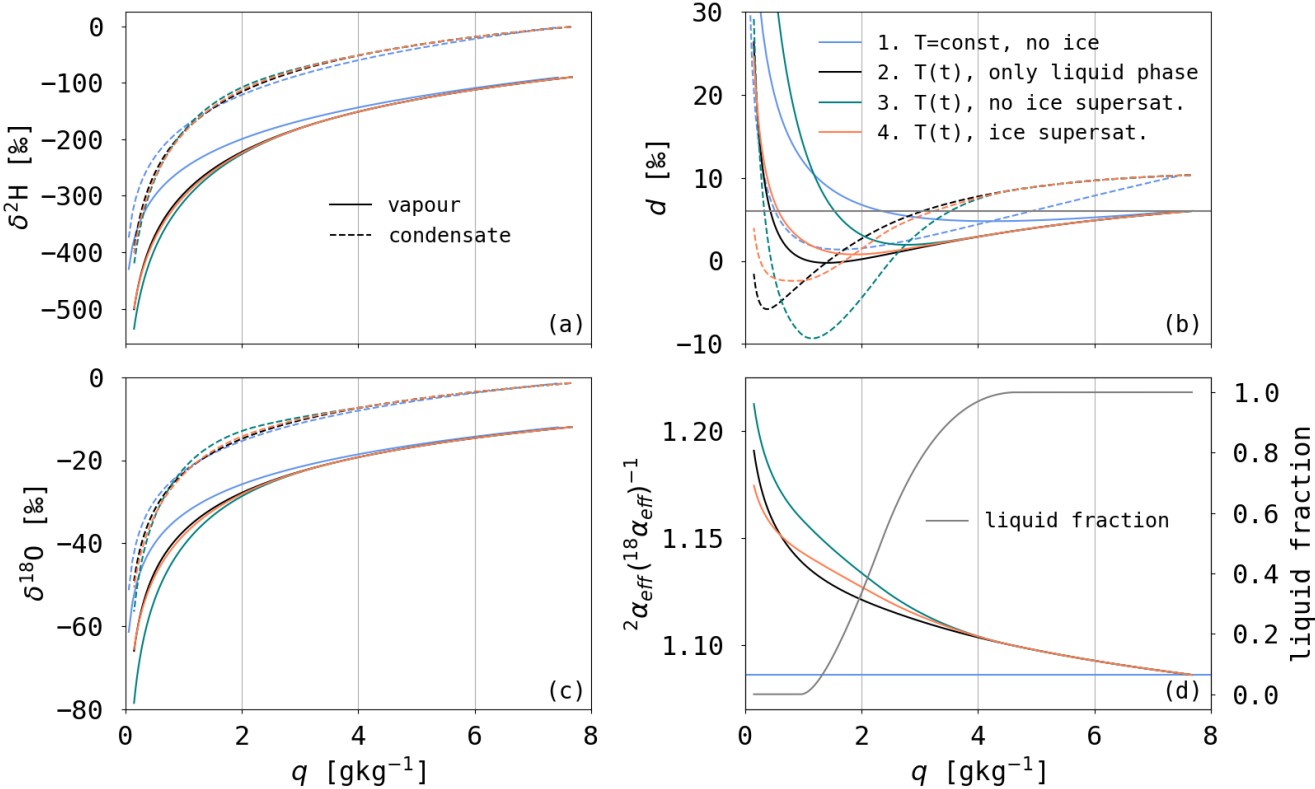

**Figure A1.** Modelled (a) $\delta^2$H, (b) $d$ and (c) $\delta^{18}$O of water vapour (solid lines) and condensate (dashed lines) versus $q$ during Rayleigh fractionation with different model setups (see text for details). Furthermore, the ratio of $^2\alpha_{eff}$ and $^{18}\alpha_{eff}$ for each model setup as a function of $q$ and the liquid fraction for the two setups with ice cloud formation is shown (d).

scenarios. These 4 scenarios show that Rayleigh fractionation during a moist adiabatic ascent can lead to a substantial decrease
of $d$ in water vapour.





*Author contributions.* HW, FA, and IT initiated the project. IT, MB, FA and HW discussed the day-to-day large-scale meteorology during/after the ACE campaign based on forecast and analyses weather charts prepared by MB. IT performed the analyses and produced the figures. KH developed and evaluated the temperature advection diagnostic during her Master thesis, with contributions from IT and FA. MW

provided the ECHAMwiso global boundary data for the regional COSMO$_{iso}$ simulations. FA, LJ, and IT designed and prepared the setup for the COSMO$_{iso}$ simulations performed during LJ's Master thesis by LJ and IT at the Swiss National Supercomputing Centre (CSCS). IG, FA and IT contributed to the production of the radiosonde data set. JG produced the rainfall data and IG the snowfall data. IT wrote the paper, supported by FA and HW. All co-authors provided feedback to the manuscript.

*Competing interests.* The authors declare that they have no conflict of interest.

*Acknowledgements.* IT acknowledges funding by the Swiss Polar Institute and Dr Frederik Paulsen. ACE was a scientific expedition carried out under the auspices of the Swiss Polar Institute, supported by funding from the ACE Foundation and Ferring Pharmaceuticals. MB acknowledges funding from the Swiss National Science Foundation (grant no. 165941) and the European Research Council 485 (ERC) under the European Union's Horizon 2020 Research and Innovation program (project INTEXseas, grant agreement no. 787652). JG received financial support from the Swiss National Science Foundation (grant no. 175700/1). IG thanks FCT/MCTES for the financial support to CESAM

(UIDP/50017/2020 and UIDB/50017/2020), through national funds, and the Swiss National Science Foundation grant PZ00P2_142684. The COSMO$_{iso}$ simulations were performed at the Swiss National Supercomputing Centre (CSCS) with the small production projects sm08 and sm32. The authors acknowledge MeteoSwiss and ECMWF for the access to the ERA-Interim reanalyses and operational forecasts. We thank Marty Ralph for co-funding the ACE radio soundings, Pascal Graf for his help during the planning of the sounding equipment and design and the ACE team for the helping hands during the launchings.





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
