# Peer review of "The role of air-sea fluxes for the water vapour isotope signals in the cold and warm sectors of extratropical cyclones over the Southern Ocean"

_Weather and Climate Dynamics, 2020_

## Referee Comment (RC1) · Anonymous Referee #1 · 27 Oct 2020

**Review of "The role of air-sea fluxes for the water vapour isotope signals in the cold and warm sectors of extra-tropical cyclones over the Southern Ocean" by Thurnherr et al.**

This study nicely combines observations and numerical modelling experiments and diagnostics to determine how the isotopic composition of water vapour is affected by air-sea fluxes induced by either cold and warm advection associated with extra-tropical cyclones. The main finding is that the cold and warm sectors have notably different stable water isotope composition. Overall, the manuscript is well written, employs appropriate methods and the conclusions are largely supported by the evidence shown in the figures. My detailed comments are below.

**Major comments**

1. This manuscript is currently quite long and covers many different data sets and time scales. Consequently, it is challenging to read. In particular, it is not clear why the climatology of cold and warm air advection from ERA-Interim needs to be included in this manuscript. This is particular the case as the climatology presented only covers December to March and therefore is not a complete climatology and as such I argue that the authors do not address their first objective "What is the occurrence frequency of cold and warm temperature advection over the Southern Ocean". Furthermore, this climatological analysis is not mentioned at all in the abstract suggesting it is not key to this manuscript. It is my opinion that this material could be removed from this manuscript.
2. Section 3 is long and the purpose of this section is not clear. This is because as well as defining the diagnostic used to identify warm and cold air advection, this section also contains an attempt to compare the COSMO simulations and the ship observations. This comparison is hard to follow and is not complete. This section would be clearer if it only covered the diagnostic and if the comparison was moved elsewhere.
3. The choice of threshold for defining cold or warm advection is not clearly explain. There is a reference to Hartmuth (2019) but some details should be included here. In particular, it is not clear why the threshold should be the same (e.g. +1K and 1K). Related to this, are these "symmetric thresholds" the reason why cold air advection is notably more common (39%) than warm air advection (12%)? Or is this difference in occurrence due to cold sectors generally being larger than warm sectors?
4. Section 2.3. More details are required here in this manuscript about how the moisture uptake is calculated. This becomes evident when a reader reaches lines 468 and 489. In both lines it is stated "the moisture uptake took place XX h before the air parcel arrived at the measurement site". This is unclear and potentially mis-leading as surely the moisture uptake does not take place at one instant in time but is an accumulation along the trajectory? Please can this be clarified / relevent detailed added to section 2.3. Furthermore, at line 463, it is not clear what is meant by "The weighted mean of the moisture source properties..." It needs to be explained how / what weighting is done. Again, please add details to section 2.3.

**Minor comments**
1. Lines 86 - 87. Please define the terms moisture sink and moisture reservoir more clearly. I find this confusing currently as in the case of evaporation from the ocean to the atmosphere, the ocean is the moisure sink (it is loosing moisture) but also could be the moisture reservoir.
2. Line 137. Related to major point 1, when it is mentioned here that ERA-Interim data will be used, it is confusing to a reader as to why.
3. Line168-169. Here the text suggests that all months are considered from ERA-Interim but elsewhere it is stated that only December-March are considered. Please clarify.

4. Line 175 - 179 and Figure 5. How does the cyclone tracking, with mean sea level pressure, work very close to or above the Antarctic coastline? In Figure 5 there are very tightly packed contours of the storm occurrence on the coastlines. Is this realistic? Please add text noting the limitations of the cyclone identification method in areas of steep terrain.
5. Line 187. The reasoning for how the boundary conditions for the COSMO simulation were created is not clear and quite confusing. It is not clear why the boundary conditions could not be taken straight from ERA-Interim at T255 resolution. It does not make sense to use ERA-Interim to drive a coarser resolution ECHAM simulations and then use that to drive the COSMO simulations. Please clarify in this manuscript why the ECHAM simulation was chosen.
6. Line 190. I believe that T106 has an equivalent resolution of 125 km, not 88 km.
7. Line 209-210. I understand why multiple limited domain simulations have been performed (computational cost) but I am concerned about how often trajectories leave the domain and also that this will not be a systematic bias i.e. there will be some times when the ship track is closer to the edge of the model domain and thus it is more likely that trajectories leave the domain. For each simulation listed in Table 1, could the number of released trajectories and the number of trajectories still in the domain after 7 (or 4?) days be added to this table? This would also give a reader a quantitative value of the number of trajectories released.
8. Line 218. How are the trajectories starting in the MBL identified? i.e how is the MBL defined? Is the BL depth taken directly from the COSMO simulations? Furthermore, why are trajectories released up to 500 hPa if only those arriving in the BL are considered?
9. Line 274. "the difference between 10 m and 24 m a.s.l air temperature is fairly small" Can this be more quantitative? I also expect this is a more valid statement for regions of cold-air advection where the MBL is well mixed, but less valid for stable MBLs. Could you utilise the data shown in Figure 8 to make this more quantatitive?
10. Line 292. If you can justify keeping the climatological aspect (and these results in section 4.1a), it needs to be more clearly explained at the start of this section why the climatology only covers December – March.
11. Line 417. Are these cases with the very high and very low observed $\Delta T_{ao}$ included in the trajectory analysis or not? Please clarify in the text.
12. Line 465. "driest point". Is this in terms of specific humidity (or relative humidity) along the trajectory? Please clarify this in the manuscript.
13. Line 521-522 and Figure 12. Are the trajectories split based on observed precipitation or the precipitation from COSMO? And more importantly, does COSMO precipitation agree with the observed precipitation? Could this panel be added to Figure 10? (ACE precipitation vs. COSMO precipitation).

**Figures Comments**
1. Figure 3: Could the sea-ice edge be made clearer in this figure? e.g. as solid contour or by hatching? Also in Figure 3b, could the warm and cold fronts be shown differently?
2. Figure 4. Can you add the dates / time period of the ship track to the caption here? It would help a reader to have this information about the time-scale at hand.
3. Figure 5. The cyclone frequency contours are hard to see (especially in Fig. 5e) – could this be improved? Maybe making one contour e.g. 30% darker?
4. Figure 12 and 13 (and in the text). The units of precipitation, and cloud water and ice are given in strange units, $mg\ kg^{-1}$. Either can these be written as mm for precipitation or $g\ kg^{-1}$ for the cloud variables, or can $mg\ kg^{-1}$ be defined in the caption / text.

---

## Referee Comment (RC2) · Anonymous Referee #2 · 4 Nov 2020

This paper analyses the impacts of air-sea fluxes on the water isotopic signal during warm and cold advection within extra-tropical cyclones. Is it based on the stable water vapour isotopic observations conducted during a ship campaign around Antarctica in the Southern Ocean sector, nicely combined with model simulations.

This article presents an excellent analysis of the dataset obtains during the ship campaign and adequately uses the numerical modeling tools to interpret these data. It is very well written. Appropriate methods are employed and the findings are largely supported by appropriate figures. The main findings reveals that different atmospheric processes occur during warm and cold advection, explaining the various stable water

vapour isotopic composition signals observed above the ocean surface.

The article is rather long and contains a lot of analyses and information. It could be lightened by removing sections about the climatology of cold, warm and zonal advection regimes, which are not absolutely necessary for the understanding of the analysis of stable water vapour isotopic composition during atmospheric transport.

The end of the stable water isotopes analysis is focusing on the interpretation of the model simulations, which is totally justified as it helps understanding the processes happening during atmospheric transport before the measurement point. However, more comparisons with the observations performed during the campaign should be provided, as these simulations originally aim at interpreting the observations.

Major comments:

L. 221: Since the identification of the moisture uptake plays an important role in the analysis, more details should be provided here about the method to identify the moisture source. What should be explained in particular is the method to calculate the transport time from the moisture source to the measurement point. Since many back-trajectories are calculated by the Lagrangian model for each ending point, this time to the moisture source but be an averaged value for all trajectories. Furthermore, each trajectory might be filled in water at different stages of the atmospheric transport, so how is the moisture source estimated for each trajectory? Is it also an average time of all the stages at which the air mass is filled in moisture?

L. 291-292: Maybe add in a few words what is exactly meant by the "occurrence frequency" and "associated air-sea moisture fluxes". Is it the frequency for each grid cell of each type of advection regime, and is the associated moisture fluxes would the mean values associated to each regime?

Section 4,3 and Fig. 12 and 13: You are often discussing variations of the median values for trajectories experiencing no surface precipitation upon arrival and for those

with surface precipitation upon arrival. On the figures, many of the differences you discuss appear to be rather small for some parameters. Contrary to the values for all trajectories, where the [25,75] percentiles are shown, these values are not presented for these two subsets. Would it be possible to include such parameters on Fig. 12 and 13? I believe this is missing when you want to legitimate the validity of the comparison of the medians when differences are very small.

Minor comments: L. 196: "realistic"

L. 246 and 254: "around 60°E" and "the cyclone at 60°E": I believe the description of the localization of the described cyclone is too vague, which makes it difficult to compare with Fig.3. Based on the Figure, I wouldn't say that the cyclone is located at 60°E but rather between 60 and 90°E. Maybe you should mark the cyclone on Fig. 3 with a different color.

L. 250: Similar remark for the 40°W cyclone, which extends over a large range of longitudes. Furthermore, it would be easier to identify it if the 40°W meridian was shown on Fig. 3. As for the previous comment, a color indicator or an arrow/letter on Fig. 3 would help identifying this specific cyclone you are describing.

L. 275: "compare orange trajectories in Fig. 4 and black dots in Fig.5" > I don't understand what should be compared here. I think you are actually refering to Fig. 7 instead of 5.

L. 361: "Fig. 10d". Check the order in which Figures are referred to and placed in the manuscript.

Fig. 11: I think it would be more readable if the xlabels where not centered on the ticks, but right-aligned, as they are rotated and they would finish at the tick.

L. 505-507: Maybe you could make a similar introduction for part c) at the end of part a). It would also be possible to modify the order of the paragraphs into a), c), b), d), in order to stay focused on the same type of events.

[Figure]

L. 519: "influenced"

In section 4.3: I find this section extremely interesting and this shows the capacity of the Lagrangian backtrajectory simulations and the regional modeling of water isotopic composition in explaining the processes affecting the water vapour within air masses during atmospheric transport. However, I think what is missing in this section is a comparison with the measurements performed on the ship. Would it be possible to include on the graphics of Fig 12 and 13 the distribution of all isotopic and meteorological values for the selected periods at the end point of the trajectories or at the surface, to see if it compares well with the COMOiso simulation and discuss such comparison in the manuscript?

Fig. 13: A legend similar to Fig. 12,h should be added to this figure.

L. 577-578: "This is indicated by the vertical d profile of non-precipitation trajectorie": I am not sure this sentence is necessary here, since it is followed by a discussion on the trajectories and the discussion on vertical profiles comes afterwards.

L. 591-593: Maybe rather place this paragraph in the discussion-conclusion, as this is not only a conclusion for 4.3.d but for the complete 4.3 section.

---

## Author Comment (AC1) · 15 Dec 2020

**wcd-2020-46**

**The role of air–sea fluxes for the water vapour isotope signals in the cold and warm sectors of extratropical cyclones over the Southern Ocean**

**Iris Thurnherr et al.**

**Replies to Review 1:**

We thank the reviewer for the constructive comments. We address below each comment point by point. Section, Figure and line numbers correspond to the first submission.

**Major comments**

1. *This manuscript is currently quite long and covers many different data sets and time scales. Consequently, it is challenging to read. In particular, it is not clear why the climatology of cold and warm air advection from ERA-Interim needs to be included in this manuscript. This is particular the case as the climatology presented only covers December to March and therefore is not a complete climatology and as such I argue that the authors do not address their first objective "What is the occurrence frequency of cold and warm temperature advection over the Southern Ocean". Furthermore, this climatological analysis is not mentioned at all in the abstract suggesting it is not key to this manuscript. It is my opinion that this material could be removed from this manuscript.*

   Answer:

   We agree that the manuscript is quite long and have shortened some lengthy parts as a consequence (e.g. in Sec. 4.1a). We however think that a short discussion of a few carefully selected climatological aspects of cold and warm advection in the Southern Ocean is needed to justify the the method and to provide a climatological context to the measurements. To keep the manuscript as short as possible, we only discuss the climatology for Dec-March, which represents the months of the measurement period. We will additionally change the following:

   - Fig. 5 (1979-2018 climatology of Dec-March) will be moved into a supplement document.
   - The ocean evaporation and precipitation composites for warm and cold advection during Dec 16 – March 17 will be added to Fig. 7 and discussed for the measurement time period.
   - We will remove the first objective as this is not a main focus of the manuscript

2. *Section 3 is long and the purpose of this section is not clear. This is because as well as defining the diagnostic used to identify warm and cold air advection, this section also contains an attempt to compare the COSMO simulations and the ship observations. This comparison is hard to follow and is not complete. This section would be clearer if it only covered the diagnostic and if the comparison was moved elsewhere.*

Answer:

We understand, that the comparison of COSMO$_{iso}$ simulations with the ship observations is not well placed in this section. We will move the comparison of measurements and model output to Sec. 4.1b and only mention it in Sec. 3, where we explain our choice of the used vertical level for air temperature.

3. *The choice of threshold for defining cold or warm advection is not clearly explain. There is a reference to Hartmuth (2019) but some details should be included here. In particular, it is not clear why the threshold should be the same (e.g. +1K and 1K). Related to this, are these "symmetric thresholds" the reason why cold air advection is notably more common (39%) than warm air advection (12%)? Or is this difference in occurrence due to cold sectors generally being larger than warm sectors?*

Answer:

In this study, we don't want to describe extreme events, but situations, in which the atmosphere and the ocean are not in thermodynamic equilibrium. Therefore, we decided to define symmetric thresholds around an isothermal near-surface stratification and do not choose the thresholds depending on the probability distribution of $\Delta T$. We added a short justification to the text in Section 3 at line 240.

But the reviewer's remark is a valid one and most probably points towards the fact that one does not a priori expect the distribution of $\Delta T$ to have a median or mode equal to 0 K. Due to the difference in heat capacity of ocean and atmosphere one may expect the mode of the $\Delta T$ distribution to be negative (see Figure 1 in this reply document). This indicates that the ocean and the atmosphere are generally not in thermodynamic equilibrium and the ocean on average loses heat to the atmosphere. As shown in Figure 1, our chosen threshold for cold advection corresponds to approximately the 50$^{th}$ percentile and the one for warm advection to the 75$^{th}$ percentile of the $\Delta T$ distribution. The reason for the different occurrence frequencies of $\Delta T>1.0°C$ *(warm temperature advection) and* $\Delta T<-1.0°C$ *(cold temperature advection)* mainly results from this observed shift of the $\Delta T$ distribution towards negative values. Accordingly, the size of the cold temperature advection regions tends to be much larger than warm temperature advection regions.

[Figure]

*Figure 1: Distribution of $T_{10m}$-SST over the ocean on the Southern hemisphere for ERA-Interim between 1979-2016 for DJF (green) and JJA (blue). The red and blue lines indicate the +/-1 °C threshold values for the warm and cold sector definition.*

4. *Section 2.3. More details are required here in this manuscript about how the moisture uptake is calculated. This becomes evident when a reader reaches lines 468 and 489. In both lines it is stated "the moisture uptake took place XX h before the air parcel arrived at the measurement site". This is unclear and potentially mis-leading as surely the moisture uptake does not take place at one instant in time but is an accumulation along the trajectory? Please can this be clarified / relevent detailed added to section 2.3. Furthermore, at line 463, it is not clear what is meant by "The weighted mean of the moisture source properties..." It needs to be explained how / what weighting is done. Again, please add details to section 2.3.*

Answer:

The moisture source diagnostic by Sodemann et al. (2008) is a well-established method and has been applied in many studies so far (see, e.g., Pfahl and Wernli 2008; Bonne et al. 2014; Aemisegger 2018). In section 2.3 at line 222, we add a short summary of the method as follows, including an explanation of the weighting along the trajectories:

"In short, this method considers the mass budget of water vapour in an air parcel. Moisture uptakes are registered whenever the specific humidity along an air parcel trajectory increases. The weight of each uptake depends on its contribution to the specific humidity of the trajectory upon arrival. If precipitation occurs (i.e. a decrease of specific humidity along the trajectory happens) after one or several uptakes, the weight of all previous uptakes is

reduced proportionally to their respective contribution to the loss. The moisture source conditions identified for each trajectory are subsequently weighted by the air parcel's specific humidity at the arrival in the boundary layer. This is done for different variables that are relevant for characterizing the moisture source conditions, such as the time of uptake, the latitude and the water vapour's isotopic composition. More details on the moisture source identification algorithm is provided in the supplement."

Furthermore, we add a more detailed description of the moisture source diagnostics in the Supplement (to avoid making the paper even longer as this was already a concern of the reviewer).

Concerning the uptake times discussed at lines 468 and 469 we now make clear that these are weighted mean times before arrival.

**Minor comments**

1. *Lines 86 - 87. Please define the terms moisture sink and moisture reservoir more clearly. I find this confusing currently as in the case of evaporation from the ocean to the atmosphere, the ocean is the moisture sink (it is losing moisture) but also could be the moisture reservoir.*

   Answer:
   We understand the reviewers concern and removed the word moisture sink. We changed the text highlighted in red: "During a diffusive process, a positive anomaly in $d$ develops in the phase towards which the flux is directed (e.g. the atmosphere during ocean evaporation), while a negative $d$ anomaly can be observed in the other, reservoir phase (e.g. rain droplets during below-cloud evaporation). If the moisture reservoir is large and well-mixed, which can e.g. be assumed for the ocean during ocean evaporation, the impact of isotopic fractionation on the isotopic composition of the reservoir can be neglected."

2. *Line 137. Related to major point 1, when it is mentioned here that ERA-Interim data will be used, it is confusing to a reader as to why.*

   Answer:
   We use ERA-interim because of its global and temporal coverage that allows to provide a climatological context to and classification of the studied summer period and to analyse the general properties of cold and warm temperature advection in terms of their associated surface evaporation flux and precipitation.

3. *Line168-169. Here the text suggests that all months are considered from ERA-Interim but elsewhere it is stated that only December-March are considered. Please clarify.*

   Answer:

We will add that only Dec-March are considered in this study.

4. *Line 175 - 179 and Figure 5. How does the cyclone tracking, with mean sea level pressure, work very close to or above the Antarctic coastline? In Figure 5 there are very tightly packed contours of the storm occurrence on the coastlines. Is this realistic? Please add text noting the limitations of the cyclone identification method in areas of steep terrain.*

   Answer:

   For the identification of cyclones, pressure minima are removed where the topography exceeds 1500 m. Therefore, a strong gradient in cyclone frequency is seen along the Antarctic coastline. We will add this information in section 2.2.1.

5. *Line 187. The reasoning for how the boundary conditions for the COSMO simulation were created is not clear and quite confusing. It is not clear why the boundary conditions could not be taken straight from ERA-Interim at T255 resolution. It does not make sense to use ERA-Interim to drive a coarser resolution ECHAM simulations and then use that to drive the COSMO simulations. Please clarify in this manuscript why the ECHAM simulation was chosen.*

   Answer:

   Since COSMOiso is an isotope-enabled regional numerical weather prediction model, it needs global isotope-enabled data for its initialization and for the boundary conditions. For consistency, ECHAM5-wiso is used for all fields. Please note that the ECHAM5-wiso simulation was nudged to ERA-Interim. We will add this information in section 2.2.2

6. *Line 190. I believe that T106 has an equivalent resolution of 125 km, not 88 km.*

   Answer:

   T106 corresponds to 1.125° horizontal resolution. This corresponds to 125 km in the meridional direction and 88 km at 45°N in the zonal direction. We will change the value to 125 km, which is independent of the latitude.

7. *Line 209-210. I understand why multiple limited domain simulations have been performed (computational cost) but I am concerned about how often trajectories leave the domain and also that this will not be a systematic bias i.e. there will be some times when the ship track is closer to the edge of the model domain and thus it is more likely that trajectories leave the domain. For each simulation listed in Table 1, could the number of released trajectories and the number of trajectories still in the domain after 7 (or 4?) days be added to this table? This would also give a reader a quantitative value of the number of trajectories released.*

   Answer:

   We understand the reviewer's concern, that the limited domains of the simulation might impact the weighted mean properties along the analysed trajectories. However, we do not think that this is a major limitation of our study, because we only consider trajectories for which a major part of the moisture at the arrival points is explained by the moisture source diagnostic (referred to as explained moisture fraction). Trajectories, which are included in this study, have an explained moisture fraction of at least 75%. We tested the sensitivity of

our results to this threshold by changing it to values between 70% and 90% and found, that the results are not sensitive to the choice of the threshold. Furthermore, trajectories, which leave the domain after a few days are not systematically neglected in this analysis (see Fig. 2 below). The excluded trajectories for a threshold of 75% stay in the model domain for 4-7 days. For the trajectories considered in the analysis, there is a larger spread of exit times from the model domain of 1-7 days. Based on this analysis, we decided to change the threshold to 70%, because with a 75% threshold most of trajectories in the simulation *leg2_run3* are excluded. We will add the number of released trajectories during warm and cold temperature advection and the percentage of these trajectories included in the analysis for each simulation in Table 1.

[Figure]

*Figure 2: Scatterplots of 1-hourly explained moisture fraction upon arrival in the MBL and the time of exit from the model domain for trajectories arriving during cold (dots) and warm (crosses) temperature advection. The colors denote the warm and cold temperature advection trajectories which are, based on a threshold of 75% for the explained moisture fraction upon arrival, included (yellow and green, respectively) and excluded (red and blue, respectively) from the analysis. The percentages in the legend denote the fraction of the total number of trajectories.*

8. *Line 218. How are the trajectories starting in the MBL identified? i.e how is the MBL defined? Is the BL depth taken directly from the COSMO simulations? Furthermore, why are trajectories released up to 500 hPa if only those arriving in the BL are considered?*

Answer:

Yes, the diagnosed boundary layer height from COSMO is used as boundary layer depth. In COSMO this is done using the bulk Richardson number. The trajectories were released up to 500 hPa, but the trajectories above the MBL are not used in this study. We will adjust the information to avoid confusion.

9. *Line 274. "the difference between 10 m and 24 m a.s.l air temperature is fairly small" Can this be more quantitative? I also expect this is a more valid statement for regions of cold-air advection where the MBL is well mixed, but less valid for stable MBLs. Could you utilise the data shown in Figure 8 to make this more quantitative?*

Answer:

Yes, we will add a more quantitative answer. Radiosonde profiles are not very precise close to the surface due to launch effects with the unwinding and oscillations of the sonde. Therefore we cannot use them for these near-surface temperature gradients. But we can give an estimate based on the simulated temperature profiles.

10. *Line 292. If you can justify keeping the climatological aspect (and these results in section 4.1a), it needs to be more clearly explained at the start of this section why the climatology only covers December – March.*

    Answer:

    Please, see our answer to major comment 1. We will state more clearly, that we focus on Dec– March to provide a climatological context to the measurement period from Dec 2016 to March 2017.

11. *Line 417. Are these cases with the very high and very low observed ΔTao included in the trajectory analysis or not? Please clarify in the text.*

    Answer:

    Yes, these events of very high and very low observed $\Delta T_{ao}$ are included in the current analysis as the identification of cold or warm temperature advection is the same in the model and measurements. We make this clearer in the text.

12. *Line 465. "driest point". Is this in terms of specific humidity (or relative humidity) along the trajectory? Please clarify this in the manuscript.*

    Answer:

    Yes, the driest point is calculated using the specific humidity. We will clarify this in the manuscript.

13. *Line 521-522 and Figure 12. Are the trajectories split based on observed precipitation or the precipitation from COSMO? And more importantly, does COSMO precipitation agree with the observed precipitation? Could this panel be added to Figure 10? (ACE precipitation vs. COSMO precipitation).*

    Answer:

    The precipitating and non-precipitating trajectories are selected based on the simulated precipitation in COSMOiso because in this section we focus on simulated SWI variables. We show the simulated and measured precipitation in Fig. 3 below, which shows the distribution of precipitation during cold and warm temperature advection for all instances when total precipitation $>$ 0mm/h along the ACE track. While the distribution of the measured precipitation shows a larger spread compared than the simulated precipitation and there is a difference in the mean and median values, as well as the 75[th] and 95[th] quartiles, overall the simulated precipitation is within the variability (and the uncertainty) of the observed values. The evaluation of precipitation in regional climate models and reanalysis

using ACE observations is part of another paper currently in preparation and is beyond the scope of this study. The reasons for the mismatch between the observed and modeled precipitation amounts include (i) needs to improve cloud/precipitation microphysical schemes in numerical weather and climate models, specifically for mixed-phase cloud representation, (ii) uncertainties in the micro-rain radar derived precipitation amounts, and (iii) discrepancies between point measurements (as provided by the radar) and the model's grid-averages. Nevertheless, overall we assume that the simulated precipitation statistics are fairly realistic. Figure 3 below shows that the simulations and measurements qualitatively agree, with clearly more precipitation during warm advection compared to cold advection. We therefore think that the analysis of the precipitating and not-precipitating trajectories is meaningful and can provide valuable information about the potential influence of precipitation-related processes on the isotopic composition of water vapour. In the future, measurements of vertical SWI profiles, especially in cloudy and precipitating situations would be most welcome to confirm the results of our study.

To keep the manuscript as short as possible, we will not add Fig. 3 to the paper. We will however make clear that we are using the simulated precipitation to create the subgroups of trajectories and we will add a statement about the comparison to the measured precipitation during ACE.

[Figure]

*Figure 3: Distributions of modeled (CI) and measured (ACE) surface precipitation during cold (blue) and warm (orange) temperature advection. The green dots show the mean values, the horizontal black lines the median values, the boxes the interquartile range and the whiskers the [5,95] percentile range of the distributions. The numbers in the x-axis labels denote the fraction of time, when it was precipitating for the corresponding advection regime and data set.*

**Figure comments**

1. *Figure 3: Could the sea-ice edge be made clearer in this figure? e.g. as solid contour or by hatching? Also in Figure 3b, could the warm and cold fronts be shown differently?*

   Answer:

Yes, we will adjust Fig. 3 to make the sea ice edge clearer and to distinguish warm and cold fronts.

2. *Figure 4. Can you add the dates / time period of the ship track to the caption here? It would help a reader to have this information about the time-scale at hand.*

   Answer:
   Yes, this will be added.

3. *Figure 5. The cyclone frequency contours are hard to see (especially in Fig. 5e) – could this be improved? Maybe making one contour e.g. 30% darker?*

   Answer:
   Yes, we will adjust this.

4. Figure 12 and 13 (and in the text). The units of precipitation, and cloud water and ice are given in strange units, mg kg$^{-1}$. Either can these be written as mm for precipitation or g kg$^{-1}$ for the cloud variables, or can mg kg$^{-1}$ be defined in the caption / text.

   Answer:
   We use mg kg$^{-1}$=10$^{-3}$ g kg$^{-1}$ for readability reasons of the tick labels. We will change the unit to 10$^{-3}$ g kg$^{-1}$ to avoid confusion.

**References:**
Aemisegger, F. and Sjolte, J.: A climatology of strong large-scale ocean evaporation events. Part II: Relevance for the deuterium excess signature of the evaporation flux, J. Clim., 31, 7313–7336, https://doi.org/10.1175/JCLI-D-17-0592.1, 2018.

Bonne, J.-L., Masson-Delmotte, V., Cattani, O., Delmotte, M., Risi, C., Sodemann, H., and Steen-Larsen, H. C.: The isotopic composition of water vapour and precipitation in Ivittuut, southern Greenland, Atmos. Chem. Phys., 14, 4419–4439, https://doi.org/10.5194/acp-14-4419-2014, 2014.

Pfahl, S., and Wernli, H.: Air parcel trajectory analysis of stable isotopes in water vapor in the eastern Mediterranean, J. Geophys. Res., 113, D20104, https://doi.org/10.1029/2008JD009839, 2008.

Sodemann, H., Schwierz, C., and Wernli, H.: Interannual variability of Greenland winter precipitation sources: Lagrangian moisture diagnostic and North Atlantic Oscillation influence, J. Geophys. Res., 113, D03 107, https://doi.org/10.1029/2007JD008503, 2008.

---

## Author Comment (AC2) · 15 Dec 2020

**wcd-2020-46**

**The role of air–sea fluxes for the water vapour isotope signals in the cold and warm sectors of extratropical cyclones over the Southern Ocean**

**Iris Thurnherr et al.**

**Replies to Review 2:**

We thank the reviewer for the constructive comments. We address below each comment point by point.

**Major comments**

1.  *L. 221: Since the identification of the moisture uptake plays an important role in the analysis, more details should be provided here about the method to identify the moisture source. What should be explained in particular is the method to calculate the transport time from the moisture source to the measurement point. Since many back-trajectories are calculated by the Lagrangian model for each ending point, this time to the moisture source but be an averaged value for all trajectories. Furthermore, each trajectory might be filled in water at different stages of the atmospheric transport, so how is the moisture source estimated for each trajectory? Is it also an average time of all the stages at which the air mass is filled in moisture?*

    Answer:
    Yes, the moisture uptake time is a weighted mean of the uptake times over all trajectories. This weighted mean uptake time is obtained in two stages. In the first stage, all the uptakes along a given trajectory are identified by following the trajectory and identifying instances, when specific humidity increases. To each of these uptakes from a given trajectory we assign a weight according to its contribution to the final specific humidity at the arrival point of the trajectory. Whenever precipitation occurs (i.e. specific humidity decreases along the trajectory), the weights of the previous uptakes are discounted accordingly. As a result from the first stage, we obtain a weighted mean time of uptake for every trajectory arriving in the boundary layer. In the second stage, we weight the resulting mean uptake time from each trajectory according to its specific humidity at the arrival point. We realise that this was not clear enough in the manuscript and added the following short explanation in Section 2.3 at line 222:

    "In short, this method considers the mass budget of water vapour in an air parcel. Moisture uptakes are registered whenever the specific humidity along an air parcel trajectory increases. The weight of each uptake depends on its contribution to the specific humidity of

the trajectory upon arrival. If precipitation occurs (i.e. a decrease of specific humidity along the trajectory happens) after one or several uptakes, the weight of all previous uptakes is reduced proportionally to their respective contribution to the loss. The moisture source conditions identified for each trajectory are subsequently weighted by the air parcel's specific humidity at the arrival in the boundary layer. This is done for different variables that are relevant for characterizing the moisture source conditions, such as the time of uptake, the latitude and the water vapour's isotopic composition. More details on the moisture source identification algorithm is provided in the supplement."

Furthermore, we add a more detailed description of the moisture source diagnostics in the Supplement (to avoid making the paper even longer as this was already a concern of the reviewer).

Concerning the uptake times discussed at lines 468 and 469 we now make clear that these are weighted mean times before arrival.

2. *L. 291-292: Maybe add in a few words what is exactly meant by the "occurrence frequency" and "associated air-sea moisture fluxes". Is it the frequency for each grid cell of each type of advection regime, and is the associated moisture fluxes would the mean values associated to each regime?*

   Answer:
   Yes, these are as you wrote. We add a few words, when we introduce these terms.

3. *Section 4,3 and Fig. 12 and 13: You are often discussing variations of the median values for trajectories experiencing no surface precipitation upon arrival and for those with surface precipitation upon arrival. On the figures, many of the differences you discuss appear to be rather small for some parameters. Contrary to the values for all trajectories, where the [25,75] percentiles are shown, these values are not presented for these two subsets. Would it be possible to include such parameters on Fig. 12 and 13? I believe this is missing when you want to legitimate the validity of the comparison of the medians when differences are very small.*

   Answer:
   By showing the two subgroups of trajectories with and without precipitation at arrival, we want to highlight that the differences in SWIs between cold and warm temperature advection are much larger than the effects due to precipitation. We find this an important and interesting result because many previous studies have shown that boundary layer water vapour isotope signals are strongly affected by precipitation (Risi et al. 2010; Aemisegger et al. 2015; Graf et al. 2019). Therefore, we show the median trajectories of these subgroups, even though the difference to all trajectories is mostly small.

[Figure]

*Figure 1: Same as Fig. 12 in the manuscript, but only showing the composites for warm temperature advection. Additionally, the [25,75] percentile range is shown for the precipitating and non-precipitating trajectories.*

We think that the figures will be overloaded if we also show the [25,75] percentiles for the non-precipitating (dry) and precipitating (wet) trajectories (see Figure 1 above, only showing the warm temperature advection trajectories). Furthermore, the [25,75] percentiles are similar for all trajectory groups (for wet, dry and all trajectories during cold and warm advection). Only for warm advection under dry conditions, there is a large difference in percentiles for the trajectory subgroups. The large percentile range for dry conditions are caused by trajectories, which have low $q$ and δ-values and arrive during leg 2 at high latitude. These are two groups of trajectories, which originate from Antarctica (see Fig. 4 in the manuscript) and are most likely warmer than the ocean surface due to large-scale subsidence. Even though these trajectories show a different dynamical history, they show the same $d$ and $E$ evolution as expected in an oversaturated environment (see Fig. 12a,d). Because (i) the [25,75] percentiles are similar for the different subgroups, (ii) the large differences in percentile range for dry warm advection trajectories are not caused by

precipitation-related processes, which we are investigating here, and (iii) the figure will be difficult to read, we will not show these ranges in Fig. 12 and 13.

We agree that the difference between the precipitating and non-precipitating trajectories is sometimes discussed in too much detail in the manuscript, without highlighting the main conclusion, which is that the largest variability in SWIs is seen due to the different temperature advection regimes. We will adjust the text in section 4.3 to highlight the main message better.

**Minor comments**

1. *L. 196: "realistic"*

   Answer:
   Thank you, we will adjust it.

2. *L. 246 and 254: "around 60° E" and "the cyclone at 60° E": I believe the description of the localization of the described cyclone is too vague, which makes it difficult to compare with Fig.3. Based on the Figure, I wouldn't say that the cyclone is located at 60° E but rather between 60 and 90° E. Maybe you should mark the cyclone on Fig. 3 with a different color.*

   Answer:
   We will adjust the description to be clearer.

3. *L. 250: Similar remark for the 40° W cyclone, which extends over a large range of longitudes. Furthermore, it would be easier to identify it if the 40° W meridian was shown on Fig. 3. As for the previous comment, a color indicator or an arrow/letter on Fig. 3 would help identifying this specific cyclone you are describing.*

   Answer:
   Also here, we will adjust the description to be clearer.

4. *L. 275: "compare orange trajectories in Fig. 4 and black dots in Fig.5" > I don't understand what should be compared here. I think you are actually refering to Fig. 7 instead of 5.*

   *Answer:*
   Thank you for pointing this out. Yes we are referring to Fig. 7 and will adjust this.

5. *L. 361: "Fig. 10d". Check the order in which Figures are referred to and placed in the manuscript.*

   Answer:
   We will check the order of the Figures.

6. *Fig. 11: I think it would be more readable if the xlabels where not centered on the ticks, but right-aligned, as they are rotated and they would finish at the tick.*

   *Answer:*
   We will adjust this.

7. *L. 505-507: Maybe you could make a similar introduction for part c) at the end of part a). It would also be possible to modify the order of the paragraphs into a), c), b), d), in order to stay focused on the same type of events.*

   Answer:
   Yes, a rearrangement of the paragraphs might help to stay focused. We will change the order and adjust the ending of paragraph a).

8. *L. 519: "influenced"*

   Answer:
   Thank you, we will adjust it.

9. *In section 4.3: I find this section extremely interesting and this shows the capacity of the Lagrangian backtrajectory simulations and the regional modeling of water isotopic composition in explaining the processes affecting the water vapour within air masses during atmospheric transport. However, I think what is missing in this section is a comparison with the measurements performed on the ship. Would it be possible to include on the graphics of Fig 12 and 13 the distribution of all isotopic and meteorological values for the selected periods at the end point of the trajectories or at the surface, to see if it compares well with the COMOiso simulation and discuss such comparison in the manuscript?*

   Answer:
   We cannot validate the Lagrangian results with our measurements as we only have measurements along the ACE track (at time=0 along the trajectories). This comparison is done in Fig. 9. As we mention in the discussion, further measurements are needed to validate our results from the trajectory analysis, for example with profile measurements, measurements in clouds or even a set of measurements with a series of instruments positioned along an air mass' pathway (which is challenging to do). We will add a sentence

at the end of the discussion to come back to the comparison of measurements and model simulations.

10. *Fig. 13: A legend similar to Fig. 12,h should be added to this figure.*

   Answer:
   Thank you, we will add this.

11. *L. 577-578: "This is indicated by the vertical d profile of non-precipitation trajectorie": I am not sure this sentence is necessary here, since it is followed by a discussion on the trajectories and the discussion on vertical profiles comes afterwards.*

   Answer:
   We will remove this sentence (and adjust the following one slightly to "The non-precipitation trajectories show a higher $d$...") as we don't want to refer to the vertical profiles at this stage.

12. *L. 591-593: Maybe rather place this paragraph in the discussion-conclusion, as this is not only a conclusion for 4.3.d but for the complete 4.3 section.*

   Answer:
   Yes, this is a conclusion for the entire Sec. 4.3. We will keep it here, but will make clear that it refers to the entire section.

References:
Aemisegger, F., Spiegel, J. K., Pfahl, S., Sodemann, H., Eugster, W., and Wernli, H.: Isotope meteorology of cold front passages: A case study combining observations and modeling, Geophys. Res. Lett., 42, 5652– 5660, doi:10.1002/2015GL063988, 2015.

Risi, C., Bony, S., Vimeux, F., Chong, M. and Descroix, L.: Evolution of the stable water isotopic composition of the rain sampled along Sahelian squall lines. Q.J.R. Meteorol. Soc., 136: 227-242. https://doi.org/10.1002/qj.485, 2010.

Graf, P., Wernli, H., Pfahl, S., and Sodemann, H.: A new interpretative framework for below-cloud effects on stable water isotopes in vapour and rain, Atmos. Chem. Phys., 19, 747–765, https://doi.org/10.5194/acp-19-747-2019, 2019.

---

## Author Response (AR1)

**wcd-2020-46**

**The role of air–sea fluxes for the water vapour isotope signals in the cold and warm sectors of extratropical cyclones over the Southern Ocean**

**Iris Thurnherr et al.**

**Replies to Review 1:**

We thank the reviewer for the constructive comments. We address below each comment point by point. Section and Figure correspond to the first submission if not stated differently. Line numbers in the comments refer to the first submission and in the answer and changes to the latexdiff document attached at the end of this document, respectively.

**Major comments**

1. *This manuscript is currently quite long and covers many different data sets and time scales. Consequently, it is challenging to read. In particular, it is not clear why the climatology of cold and warm air advection from ERA-Interim needs to be included in this manuscript. This is particular the case as the climatology presented only covers December to March and therefore is not a complete climatology and as such I argue that the authors do not address their first objective "What is the occurrence frequency of cold and warm temperature advection over the Southern Ocean". Furthermore, this climatological analysis is not mentioned at all in the abstract suggesting it is not key to this manuscript. It is my opinion that this material could be removed from this manuscript.*

   Answer:
   We agree that the manuscript is quite long and have shortened some lengthy parts as a consequence (e.g. in Sec. 4.1a). We however think that a short discussion of a few carefully selected climatological aspects of cold and warm advection in the Southern Ocean is needed to justify the method and to provide a climatological context to the measurements.

   Changes:
   To keep the manuscript as short as possible, we only discuss the climatology for Dec-March, which represents the months of the measurement period. We additionally changed the following:

   - Fig. 5 (1979-2018 climatology of Dec-March) is moved into a supplement document.
   - The ocean evaporation and precipitation composites for warm and cold advection during Dec 16 – March 17 is added to Fig. 7 and discussed for the measurement time period: see Fig.5 in revised manuscript and Section 4.1, lines 315 - 381.
   - We removed the first objective as this is not a main focus of the manuscript: see lines 124-130.

2. *Section 3 is long and the purpose of this section is not clear. This is because as well as defining the diagnostic used to identify warm and cold air advection, this section also contains an*

*attempt to compare the COSMO simulations and the ship observations. This comparison is hard to follow and is not complete. This section would be clearer if it only covered the diagnostic and if the comparison was moved elsewhere.*

Answer:

We understand, that the comparison of COSMO$_{iso}$ simulations with the ship observations is not well placed in this section.

Changes:

We moved the comparison of measurements and model output to Sec. 4.1b in the revised manuscript (lines 391-397) and only mention it in Sec. 3, where we explain our choice of the used vertical level for air temperature (lines 305-306 ):

lines 391-397: "The largest difference in the identification of cold and warm temperature advection between the measurements and COSMO$_{iso}$ is observed during leg 2 (compare orange trajectories in Fig. 4 and black dots in Fig. 5). Two warm temperature advection events are identified during leg 2 in COSMO$_{iso,}$ which were categorised as zonal flow using the measurements. During these two events, air is advected northwards from Antarctica towards the ship's position. The large positive $\Delta T_{ao}$ in COSMO$_{iso}$ could be caused by adiabatic warming during the descent in a katabatic wind event. These two warm temperature advection events thus differ from a typical warm temperature advection event as generally observed along the ACE track in the warm sector of an extratropical cyclone."

lines 305-306: "Overall, the identified cold and warm temperature advection events using COSMO$_{iso}$ agree well with the measurements and represent similar environmental conditions (see also Sect. 4.2)."

3. *The choice of threshold for defining cold or warm advection is not clearly explain. There is a reference to Hartmuth (2019) but some details should be included here. In particular, it is not clear why the threshold should be the same (e.g. +1K and 1K). Related to this, are these "symmetric thresholds" the reason why cold air advection is notably more common (39%) than warm air advection (12%)? Or is this difference in occurrence due to cold sectors generally being larger than warm sectors?*

Answer:

In this study, we don't want to describe extreme events, but situations, in which the atmosphere and the ocean are not in thermodynamic equilibrium. Therefore, we decided to define symmetric thresholds around an isothermal near-surface stratification and do not choose the thresholds depending on the probability distribution of $\Delta T$.

But the reviewer's remark is a valid one and most probably points towards the fact that one does not a priori expect the distribution of $\Delta T$ to have a median or mode equal to 0 K. Due to the difference in heat capacity of ocean and atmosphere one may expect the mode of the $\Delta T$ distribution to be negative (see Figure 1 in this reply document). This indicates that the

ocean and the atmosphere are generally not in thermodynamic equilibrium and the ocean on average loses heat to the atmosphere. As shown in Figure 1, our chosen threshold for cold advection corresponds to approximately the 50[th] percentile and the one for warm advection to the 75[th] percentile of the $\Delta T$ distribution. The reason for the different occurrence frequencies of $\Delta T > 1.0°C$ *(warm temperature advection)* and $\Delta T < -1.0°C$ *(cold temperature advection)* mainly results from this observed shift of the $\Delta T$ distribution towards negative values. Accordingly, the size of the cold temperature advection regions tends to be much larger than warm temperature advection regions.

Changes:
We added a short justification of the chosen thresholds to the text in Section 3 at line 253-255:
"In this study, we analyse situations, in which the atmosphere and the ocean are not in thermodynamic equilibrium. Therefore, symmetric thresholds around an isothermal near-surface stratification are chosen."

[Figure]

*Figure 1: Distribution of $T_{10m}$-SST over the ocean on the Southern hemisphere for ERA-Interim between 1979-2016 for DJF (green) and JJA (blue). The red and blue lines indicate the +/-1 °C threshold values for the warm and cold sector definition.*

4. *Section 2.3. More details are required here in this manuscript about how the moisture uptake is calculated. This becomes evident when a reader reaches lines 468 and 489. In both lines it is stated "the moisture uptake took place XX h before the air parcel arrived at the measurement*

*site". This is unclear and potentially mis-leading as surely the moisture uptake does not take place at one instant in time but is an accumulation along the trajectory? Please can this be clarified / relevent detailed added to section 2.3. Furthermore, at line 463, it is not clear what is meant by "The weighted mean of the moisture source properties..." It needs to be explained how / what weighting is done. Again, please add details to section 2.3.*

Answer and changes:

The moisture source diagnostic by Sodemann et al. (2008) is a well-established method and has been applied in many studies so far (see, e.g., Pfahl and Wernli 2008; Bonne et al. 2014; Aemisegger 2018). In section 2.3, we added a short summary of the method as follows, including an explanation of the weighting along the trajectories (lines 229-237):

"In short, this method considers the mass budget of water vapour in an air parcel. Moisture uptakes are registered whenever the specific humidity along an air parcel trajectory increases. The weight of each uptake depends on its contribution to the specific humidity of the trajectory upon arrival. If precipitation occurs (i.e. a decrease of specific humidity along the trajectory happens) after one or several uptakes, the weight of all previous uptakes is reduced proportionally to their respective contribution to the loss. The moisture source conditions identified for each trajectory are subsequently weighted by the air parcel's specific humidity at the arrival in the boundary layer. This is done for different variables that are relevant for characterizing the moisture source conditions, such as the time of uptake, the latitude and the water vapour's isotopic composition. More details on the moisture source identification algorithm is provided in the supplement."

Furthermore, we added a more detailed description of the moisture source diagnostics in the Supplement (to avoid making the paper even longer as this was already a concern of the reviewer).

Concerning the uptake times discussed at lines 468 and 469 (in the submitted manuscript) we now make clear that these are weighted mean times before arrival: see lines 523 and 614.

**Minor comments**

1. *Lines 86 - 87. Please define the terms moisture sink and moisture reservoir more clearly. I find this confusing currently as in the case of evaporation from the ocean to the atmosphere, the ocean is the moisture sink (it is losing moisture) but also could be the moisture reservoir.*

   Answer:
   We understand the reviewers concern and removed the word moisture sink.

   Changes:

We changed the text as follows (lines 86-90): "During a diffusive process, a positive anomaly in *d* develops in the phase towards which the flux is directed (e.g. the atmosphere during ocean evaporation), while a negative *d* anomaly can be observed in the other, reservoir phase (e.g. rain droplets during below-cloud evaporation). If the moisture reservoir is large and well-mixed, which can be assumed for the ocean during ocean evaporation, the impact of isotopic fractionation on the isotopic composition of the reservoir can be neglected."

2. *Line 137. Related to major point 1, when it is mentioned here that ERA-Interim data will be used, it is confusing to a reader as to why.*

   Answer:
   We use ERA-interim because of its global and temporal coverage that allows to provide a climatological context to and classification of the studied summer period and to analyse the general properties of cold and warm temperature advection in terms of their associated surface evaporation flux and precipitation.

3. *Line168-169. Here the text suggests that all months are considered from ERA-Interim but elsewhere it is stated that only December-March are considered. Please clarify.*

   Answer and changes:
   We added that only Dec-March are considered in this study (lines 171-172):
   "Six-hourly data from the ERA-Interim reanalyses (Dee et al., 2011) from December 2016 to March 2017 are used..."

4. *Line 175 - 179 and Figure 5. How does the cyclone tracking, with mean sea level pressure, work very close to or above the Antarctic coastline? In Figure 5 there are very tightly packed contours of the storm occurrence on the coastlines. Is this realistic? Please add text noting the limitations of the cyclone identification method in areas of steep terrain.*

   Answer:
   For the identification of cyclones, pressure minima are removed where the topography exceeds 1500 m. Therefore, a strong gradient in cyclone frequency is seen along the Antarctic coastline.

   Changes:
   We added this information in section 2.2.1 (lines 180-181):
   "For the identification of the cyclones, pressure minima are removed where the topography exceeds 1500 m."

5. *Line 187. The reasoning for how the boundary conditions for the COSMO simulation were created is not clear and quite confusing. It is not clear why the boundary conditions could not be taken straight from ERA-Interim at T255 resolution. It does not make sense to use ERA-Interim to drive a coarser resolution ECHAM simulations and then use that to drive the COSMO simulations. Please clarify in this manuscript why the ECHAM simulation was chosen.*

Answer:

Since COSMOiso is an isotope-enabled regional numerical weather prediction model, it needs global isotope-enabled data for its initialization and for the boundary conditions. For consistency, ECHAM5-wiso is used for all fields. Please note that the ECHAM5-wiso simulation was nudged to ERA-Interim.

Changes:

We added this information in section 2.2.2 (lines 191-192):

"Since COSMO$_{iso}$ is an isotope-enabled regional numerical weather prediction model, global SWI data are needed for its initialization and for the boundary conditions."

6. *Line 190. I believe that T106 has an equivalent resolution of 125 km, not 88 km.*

Answer:

T106 corresponds to 1.125° horizontal resolution. This corresponds to 125 km in the meridional direction and 88 km at 45°N in the zonal direction.

Changes:

We changed the value to 125 km, which is independent of the latitude: see line 197.

7. *Line 209-210. I understand why multiple limited domain simulations have been performed (computational cost) but I am concerned about how often trajectories leave the domain and also that this will not be a systematic bias i.e. there will be some times when the ship track is closer to the edge of the model domain and thus it is more likely that trajectories leave the domain. For each simulation listed in Table 1, could the number of released trajectories and the number of trajectories still in the domain after 7 (or 4?) days be added to this table? This would also give a reader a quantitative value of the number of trajectories released.*

Answer:

We understand the reviewer's concern, that the limited domains of the simulation might impact the weighted mean properties along the analysed trajectories. However, we do not think that this is a major limitation of our study, because we only consider trajectories for which a major part of the moisture at the arrival points is explained by the moisture source diagnostic (referred to as explained moisture fraction). Trajectories, which are included in this study, have an explained moisture fraction of at least 75%. We tested the sensitivity of our results to this threshold by changing it to values between 70% and 90% and found, that the results are not sensitive to the choice of the threshold. Furthermore, trajectories, which leave the domain after a few days are not systematically neglected in this analysis (see Fig. 2 below). The excluded trajectories for a threshold of 75% stay in the model domain for 4-7 days. For the trajectories considered in the analysis, there is a larger spread of exit times from the model domain of 1-7 days.

Changes:

Based on this analysis, we decided to change the threshold to 70%, because with a 75% threshold most of trajectories in the simulation *leg2_run3* are excluded.
We added the number of released trajectories during warm and cold temperature advection and the percentage of these trajectories included in the analysis for each simulation in Table 1 (page 9 in latexdiff document).

[Figure]

*Figure 2: Scatterplots of 1-hourly explained moisture fraction upon arrival in the MBL and the time of exit from the model domain for trajectories arriving during cold (dots) and warm (crosses) temperature advection. The colors denote the warm and cold temperature advection trajectories which are, based on a threshold of 75% for the explained moisture fraction upon arrival, included (yellow and green, respectively) and excluded (red and blue, respectively) from the analysis. The percentages in the legend denote the fraction of the total number of trajectories.*

8. *Line 218. How are the trajectories starting in the MBL identified? i.e how is the MBL defined? Is the BL depth taken directly from the COSMO simulations? Furthermore, why are trajectories released up to 500 hPa if only those arriving in the BL are considered?*

Answer:

Yes, the diagnosed boundary layer height from COSMO is used as boundary layer depth. In COSMO this is done using the bulk Richardson number. The trajectories were released up to 500 hPa, but the trajectories above the MBL are not used in this study.

Changes:

We adjusted the information to avoid confusion(lines 222-223):

"...at pressure levels in 10 hPa-steps between the surface and the MBL top as identified by COSMO$_{iso}$".

9. *Line 274. "the difference between 10 m and 24 m a.s.l air temperature is fairly small" Can this be more quantitative? I also expect this is a more valid statement for regions of cold-air advection where the MBL is well mixed, but less valid for stable MBLs. Could you utilise the data shown in Figure 8 to make this more quantitative?*

Answer:

Yes, we will add a more quantitative answer. Radiosonde profiles are not very precise close to the surface due to launch effects with the unwinding and oscillations of the sonde. Therefore we cannot use them for these near-surface temperature gradients. But we can give an estimate based on the simulated temperature profiles.

Changes:

We added a more quantitative answer on lines 289-306:

"The higher level of air temperature used for the calculation of the measured advection events leads to slightly higher frequencies in cold temperature advection and lower frequencies in warm temperature advection compared to the $COSMO_{iso}$ and ERA-Interim advection events, because lower air temperature is expected at higher altitude. In $COSMO_{iso}$, the median temperature difference between the lowest model level at 10m a.s.l. and the second lowest model level at 35m a.s.l. is 0.25 [0.19,0.29] °C (the brackets denote the [25,75] percentile range). Using the temperature at 35m a.s.l. instead of 10m a.s.l leads to an increase in total hours of cold temperature advection by 13% and a decrease of warm temperature advection by 9% along the ACE track. Nonetheless, the difference between 10m and 24m a.s.l. air temperature is fairly small and the advection frequencies in $COSMO_{iso}$ and ERA-Interim are similar to the advection frequencies in the measurements. The choice of air temperature altitude mainly changes the length of the advection events by a few hours. Overall, the identified cold and warm temperature advection events using $COSMO_{iso}$ agree well with the measurements and represent similar environmental conditions (see also Sect. 4.2)."

10. *Line 292. If you can justify keeping the climatological aspect (and these results in section 4.1a), it needs to be more clearly explained at the start of this section why the climatology only covers December – March.*

    Answer:

    Please, see our answer and changes for major comment 1. We stated more clearly, that we focus on Dec– March to provide a climatological context to the measurement period from Dec 2016 to March 2017.

11. *Line 417. Are these cases with the very high and very low observed ΔTao included in the trajectory analysis or not? Please clarify in the text.*

    Answer:

    Yes, these events of very high and very low observed $\Delta T_{ao}$ are included in the current analysis as the identification of cold or warm temperature advection is the same in the model and measurements.

    Changes:

We made this clearer in the text by adding "These events are included in the analysis as the identification of cold or warm temperature advection agrees in the model and the measurements." on lines 469-470.

12. *Line 465. "driest point". Is this in terms of specific humidity (or relative humidity) along the trajectory? Please clarify this in the manuscript.*

Answer:

Yes, the driest point is calculated using the specific humidity.

Changes:

We clarified this in the manuscript by adding "in terms of specific humidity" on line 520.

13. *Line 521-522 and Figure 12. Are the trajectories split based on observed precipitation or the precipitation from COSMO? And more importantly, does COSMO precipitation agree with the observed precipitation? Could this panel be added to Figure 10? (ACE precipitation vs. COSMO precipitation).*

Answer:

The precipitating and non-precipitating trajectories are selected based on the simulated precipitation in COSMOiso because in this section we focus on simulated SWI variables. We show the simulated and measured precipitation in Fig. 3 below, which shows the distribution of precipitation during cold and warm temperature advection for all instances when total precipitation $> 0$mm/h along the ACE track. While the distribution of the measured precipitation shows a larger spread compared than the simulated precipitation and there is a difference in the mean and median values, as well as the $75^{th}$ and $95^{th}$ quartiles, overall the simulated precipitation is within the variability (and the uncertainty) of the observed values. The evaluation of precipitation in regional climate models and reanalysis using ACE observations is part of another paper currently in preparation and is beyond the scope of this study. The reasons for the mismatch between the observed and modeled precipitation amounts include (i) needs to improve cloud/precipitation microphysical schemes in numerical weather and climate models, specifically for mixed-phase cloud representation, (ii) uncertainties in the micro-rain radar derived precipitation amounts, and (iii) discrepancies between point measurements (as provided by the radar) and the model's grid-averages. Nevertheless, overall we assume that the simulated precipitation statistics are fairly realistic. Figure 3 below shows that the simulations and measurements qualitatively agree, with clearly more precipitation during warm advection compared to cold advection. We therefore think that the analysis of the precipitating and not-precipitating trajectories is meaningful and can provide valuable information about the potential influence of precipitation-related processes on the isotopic composition of water vapour. In the future, measurements of vertical SWI profiles, especially in cloudy and precipitating situations would be most welcome to confirm the results of our study.

Changes:

To keep the manuscript as short as possible, we did not add Fig. 3 of this document to the paper. We however made clear that we are using the simulated precipitation to create the subgroups of trajectories (lines 577-580) and we added a statement about the comparison to the measured precipitation during ACE (lines 422-428).

lines 577-580: "To analyse the effect of precipitation on the water vapour isotopic composition of the air parcels, the median properties along precipitating (surface preciptiation $>0.01$ mm h$^{-1}$ ) and non-precipitating (surface preciptiation $\leq 0.01$ mm h$^{-1}$ trajectories upon arrival are calculated based on the simulated surface precipitation in COSMO$_{iso}$."

lines 422-428: "Surface precipitation in ERA-Interim (Fig. 5b,d) and COSMO$_{iso}$ (not shown) shows larger median values for cold and warm advection, but qualitatively agrees with the observed rainfall intensities with typically heavier precipitation during warm than cold temperature advection. The reasons for the mismatch between the observed and modeled precipitation amounts include (i) limitations of the microphysical scheme used in numerical weather models, specifically for mixed-phase clouds, (ii) uncertainties in the micro-rain radar derived precipitation amounts, and (iii) discrepancies between point measurements (as provided by the radar) and the model's grid-averages. Nevertheless, overall we assume that the simulated precipitation statistics are fairly realistic."

[Figure]

*Figure 3: Distributions of modeled (CI) and measured (ACE) surface precipitation during cold (blue) and warm (orange) temperature advection. The green dots show the mean values, the horizontal black lines the median values, the boxes the interquartile range and the whiskers the [5,95] percentile range of the distributions. The numbers in the x-axis labels denote the fraction of time, when it was precipitating for the corresponding advection regime and data set.*

**Figure comments**

1. *Figure 3: Could the sea-ice edge be made clearer in this figure? e.g. as solid contour or by hatching? Also in Figure 3b, could the warm and cold fronts be shown differently?*

Answer and changes:

Yes, we adjusted Fig. 3 to make the sea ice edge clearer and to distinguish warm and cold fronts.

2. *Figure 4. Can you add the dates / time period of the ship track to the caption here? It would help a reader to have this information about the time-scale at hand.*

Answer and changes:

Yes, this is added.

3. *Figure 5. The cyclone frequency contours are hard to see (especially in Fig. 5e) – could this be improved? Maybe making one contour e.g. 30% darker?*

Answer and changes:

Yes, we adjusted this.

4. Figure 12 and 13 (and in the text). The units of precipitation, and cloud water and ice are given in strange units, mg kg$^{-1}$. Either can these be written as mm for precipitation or g kg$^{-1}$ for the cloud variables, or can mg kg$^{-1}$ be defined in the caption / text.

Answer and changes:

We use mg kg$^{-1}$=10$^{-3}$ g kg$^{-1}$ for readability reasons of the tick labels. We changed the unit to 10$^{-3}$ g kg$^{-1}$ to avoid confusion: see Fig. 11 and 12 in the latexdiff document.

*you want to legitimate the validity of the comparison of the medians when differences are very small.*

Answer:

By showing the two subgroups of trajectories with and without precipitation at arrival, we want to highlight that the differences in SWIs between cold and warm temperature advection are much larger than the effects due to precipitation. We find this an important and interesting result because many previous studies have shown that boundary layer water vapour isotope signals are strongly affected by precipitation (Risi et al. 2010; Aemisegger et al. 2015; Graf et al. 2019). Therefore, we show the median trajectories of these subgroups, even though the difference to all trajectories is mostly small.

[Figure]

*Figure 1: Same as Fig. 12 in the manuscript, but only showing the composites for warm temperature advection. Additionally, the [25,75] percentile range is shown for the precipitating and non-precipitating trajectories.*

We think that the figures will be overloaded if we also show the [25,75] percentiles for the non-precipitating (dry) and precipitating (wet) trajectories (see Figure 1 above, only showing the warm temperature advection trajectories). Furthermore, the [25,75] percentiles

are similar for all trajectory groups (for wet, dry and all trajectories during cold and warm advection). Only for warm advection under dry conditions, there is a large difference in percentiles for the trajectory subgroups. The large percentile range for dry conditions are caused by trajectories, which have low $q$ and $\delta$-values and arrive during leg 2 at high latitude. These are two groups of trajectories, which originate from Antarctica (see Fig. 4 in the manuscript) and are most likely warmer than the ocean surface due to large-scale subsidence. Even though these trajectories show a different dynamical history, they show the same $d$ and $E$ evolution as expected in an oversaturated environment (see Fig. 12a,d). Because (i) the [25,75] percentiles are similar for the different subgroups, (ii) the large differences in percentile range for dry warm advection trajectories are not caused by precipitation-related processes, which we are investigating here, and (iii) the figure will be difficult to read, we will not show these ranges in Fig. 12 and 13.

Changes:
We agree that the difference between the precipitating and non-precipitating trajectories is sometimes discussed in too much detail in the manuscript, without highlighting the main conclusion, which is that the largest variability in SWIs is seen due to the different temperature advection regimes. We adjusted the text in section 4.3 to highlight the main message better, specifically on lines 575-590, 595 and 662-669.

**Minor comments**

1. *L. 196: "realistic"*

   Answer and changes:
   Thank you, we adjusted it (line 202).

2. *L. 246 and 254: "around 60 ▫ E" and "the cyclone at 60 ▫ E": I believe the description of the localization of the described cyclone is too vague, which makes it difficult to compare with Fig.3. Based on the Figure, I wouldn't say that the cyclone is located at 60 ▫ E but rather between 60 and 90 ▫ E. Maybe you should mark the cyclone on Fig. 3 with a different color.*

   Answer and changes:
   We adjusted the description to be clearer (lines 263 and 271).

3. *L. 250: Similar remark for the 40 ▫ W cyclone, which extends over a large range of longitudes. Furthermore, it would be easier to identify it if the 40 ▫ W meridian was shown on Fig. 3. As for the previous comment, a color indicator or an arrow/letter on Fig. 3 would help identifying this specific cyclone you are describing.*

Answer and changes:

Also here, we adjusted the description to be clearer (line 268 and 270).

4. *L. 275: "compare orange trajectories in Fig. 4 and black dots in Fig.5" > I don't understand what should be compared here. I think you are actually refering to Fig. 7 instead of 5.*

   *Answer and changes:*
   Thank you for pointing this out. Yes, we are referring to Fig. 7 and adjusted this (line 392).

5. *L. 361: "Fig. 10d". Check the order in which Figures are referred to and placed in the manuscript.*

   Answer and changes:
   We checked the order of the Figures and moved Fig.10 to position 7 in the revised manuscript.

6. *Fig. 11: I think it would be more readable if the xlabels where not centered on the ticks, but right-aligned, as they are rotated and they would finish at the tick.*

   *Answer and changes:*
   We adjusted this.

7. *L. 505-507: Maybe you could make a similar introduction for part c) at the end of part a). It would also be possible to modify the order of the paragraphs into a), c), b), d), in order to stay focused on the same type of events.*

   Answer and changes:
   Yes, a rearrangement of the paragraphs might help to stay focused. We changed the order as suggested and adjusted the ending of paragraph 4.3a) and 4.3c).
   (lines 560-562): "To better understand these changes between the isotopic composition at the moisture source and the point of measurement, the temporal evolution of the isotopic composition and other environmental variables are analysed along the backward trajectories for the 4 days before arrival."
   (lines 630-632): "Analogously to the cold temperature advection analysis, the temporal evolution of the isotopic composition and other environmental variables along the 4-day backward trajectories are analysed in the following to identify the main process affecting SWIs in the warm sector."

8.  L. 519: "influenced"

    Answer and changes:
    Thank you, we adjusted it.

9.  In section 4.3: I find this section extremely interesting and this shows the capacity of the
    Lagrangian backtrajectory simulations and the regional modeling of water isotopic
    composition in explaining the processes affecting the water vapour within air masses during
    atmospheric transport. However, I think what is missing in this section is a comparison with
    the measurements performed on the ship. Would it be possible to include on the graphics
    of Fig 12 and 13 the distribution of all isotopic and meteorological values for the selected
    periods at the end point of the trajectories or at the surface, to see if it compares well with
    the COMOiso simulation and discuss such comparison in the manuscript?

    Answer:
    We cannot validate the Lagrangian results with our measurements as we only have
    measurements along the ACE track (at time=0 along the trajectories). This comparison is
    done in Fig. 9. As we mention in the discussion, further measurements are needed to
    validate our results from the trajectory analysis, for example with profile measurements,
    measurements in clouds or even a set of measurements with a series of instruments
    positioned along an air mass' pathway (which is challenging to do).

    Changes:
    We added two sentences at the end of 4.3d) to come back to the comparison of
    measurements and model simulations (lines 684-689):
    "The ACE measurements confirmed that the SWI signals in the lower MBL along the ACE
    ship track were adequately simulated in $COSMO_{iso}$, but cannot be used to verify the
    Lagrangian analysis. The importance of the described processes, such as cloud formation
    and below-cloud processes, during moisture transport needs more detailed future studies
    which include the comparison of measurements and model data in the upper MBL and
    campaigns including upstream measurements of the air mass properties."

10. Fig. 13: A legend similar to Fig. 12,h should be added to this figure.

    Answer and changes:
    Thank you, we added this.

11. L. 577-578: "This is indicated by the vertical d profile of non-precipitation trajectorie": I am
    not sure this sentence is necessary here, since it is followed by a discussion on the
    trajectories and the discussion on vertical profiles comes afterwards.

Answer and changes:

We removed this sentence as we don't want to refer to the vertical profiles at this stage.

12. *L. 591-593: Maybe rather place this paragraph in the discussion-conclusion, as this is not only a conclusion for 4.3.d but for the complete 4.3 section.*

Answer and changes:

Yes, this is a conclusion for the entire Sec. 4.3. We keep it here, but made clear that it refers to the entire section (line 678-680):

"In summary, the analysis..."

References:

[revised manuscript text omitted]